# Priming with LSD1 inhibitors promotes the persistence and antitumor effect of adoptively transferred T cells

Fengqi Qiu [1,4], Peishan Jiang[1,2,4], Guiheng Zhang[1,2], Jie An[2], Kexin Ruan[1], Xiaowen Lyu [3] ✉, Jianya Zhou [1] ✉ & Wanqiang Sheng [1,2] ✉

The antitumor efficacy of adoptively transferred T cells is limited by their poor persistence, in part due to exhaustion, but the underlying mechanisms and potential interventions remain underexplored. Here, we show that targeting histone demethylase LSD1 by chemical inhibitors reshapes the epigenome of in vitro activated and expanded CD8+ T cells, and potentiates their antitumor efficacy. Upon T cell receptor activation and IL-2 signaling, a timely and transient inhibition of LSD1 suffices to improve the memory phenotype of mouse CD8+ T cells, associated with a better ability to produce multiple cytokines, resist exhaustion, and persist in both antigen-dependent and -independent manners after adoptive transfer. Consequently, OT1 cells primed with LSD1 inhibitors demonstrate an enhanced antitumor effect in OVA-expressing solid tumor models implanted in female mice, both as a standalone treatment and in combination with PD-1 blockade. Moreover, priming with LSD1 inhibitors promotes polyfunctionality of human CD8+ T cells, and increases the persistence and antitumor efficacy of human CD19-CAR T cells in both leukemia and solid tumor models. Thus, pharmacological inhibition of LSD1 could be exploited to improve adoptive T cell therapy.

Adoptive transfer of antigen receptor-engineered T cells is an emerging therapeutic approach in cancer treatment. Chimeric antigen receptor (CAR) T-cell therapy, particularly in B cell malignancies, has shown high objective responsive rates[1–3]. However, over half of the patients experience relapse in CAR T therapy. In solid tumors, the efficacy of CAR T therapy has been limited, faced with challenges such as lack of suitable antigens, tumor heterogeneity, and the immunosuppressive tumor microenvironment (TME)[4]. T-cell receptor-engineered T (TCR T) cell therapy is being explored as an alternative for solid tumors and has already shown promising outcomes, but tumor relapse remains an issue[5]. The limited persistence of both CAR T and TCR T cells, in part due to T-cell exhaustion, is a significant obstacle impeding their long-lasting therapeutic effect. Thus, it is an unmet need to overcome exhaustion and promote the persistence of adoptively transferred T cells.

T-cell exhaustion is induced by persistent TCR stimulation together with a variety of microenvironmental signals, for instance, TGFβ, IL-10, PGE2, and adenosine. It is characterized by a progressive loss of effector functions, poor persistence and proliferation capacity, and high-level expression of inhibitory receptors[6]. The phenotypic changes in exhausted T cells (Tex) are linked to extensive remodeling of chromatin landscapes, including changes in DNA methylation, histone modifications, and chromatin accessibility, which differ significantly from those in effector (Teff) and memory (Tmem) T cells[7–12]. It has become evident that the exhaustion-associated chromatin landscapes are stable and fixed, as current therapeutic approaches for relieving

[1]Department of Respiratory Disease, Thoracic Disease Center, The First Affiliated Hospital, Zhejiang University School of Medicine, Hangzhou, China. [2]Institute of Immunology and Liangzhu Laboratory, Zhejiang University School of Medicine, Hangzhou, China. [3]State Key Laboratory of Cellular Stress Biology, Fujian Provincial Key Laboratory of Reproductive Health Research, School of Medicine, Xiamen University, Xiamen, China. [4]These authors contributed equally: Fengqi Qiu, Peishan Jiang. ✉e-mail: xiaowenlyu@xmu.edu.cn; zhoujy@zju.edu.cn; wanqiang_sheng@zju.edu.cn

T-cell exhaustion, for instance by PD-1 blockade, lack the ability to revert them[7,13]. This "epigenetic barrier" is thought to prevent bona fide reinvigoration of terminal Tex cells. Considering their pivotal role in rewiring and maintaining chromatin landscapes, perturbing epigenetic regulators may offer a way to change or even revert the exhaustion-associated chromatin state in Tex cells. While direct conversion of terminal Tex cells back to effector or memory T cells is not well evidenced, recent studies suggest that targeting specific epigenetic regulators could prevent the acquisition of exhaustion by disrupting the establishment of associated chromatin landscapes. For instance, blocking de novo DNA methylation by depleting *Dnmt3a* in activated T cells helps avert exhaustion-associated methylation patterns, thus preserving the proliferative capacity and effector functions of both endogenous CD8+ T cells and adoptively transferred EphA2-CAR T cells[14,15]. Similarly, treating CAR T cells with very low-dose (10 nM) decitabine, a DNA hypomethylating agent, has been reported to enhance the persistence and effector functions of human CD19-CAR T cells both in vitro and in animal models[16]. However, in another study, a moderate increase in the dose of decitabine (100 nM) turns to decrease the proliferative capacity of unmodified human CD8+ T cells[17]. Additionally, depletion of TET2, a member of the TET family responsible for actively removing the methyl group from 5-methylcytosine (5mC) on genomic DNA, results in superior persistence and proliferation of human CD19-CAR T cells[18,19]. These findings emphasize the importance of understanding the complex roles of DNA methylation in T-cell functionality. Histone modifications, involving diverse chemical moieties and distinct modification sites, are central to chromatin landscapes, but their roles and mechanisms of action in T-cell exhaustion are not fully understood. H3K27me3, catalyzed by the methyltransferase EZH2, has recently been implicated in the functional reinvigoration of GD2-CAR T cells[20]. Additionally, in two recent studies, disruption of H3K9me3 by depleting SUV39H1 has been reported to enhance CAR T-cell persistence and antitumor effect in solid tumor models[21,22]. A deeper understanding of the chromatin landscapes and their regulatory mechanisms could open new therapeutic opportunities to combat the exhaustion of adoptively transferred T cells.

Our recent study reported that depletion of H3K4me1/2 demethylase LSD1 increased intratumoral persistence of the progenitor-exhausted T cells and their proliferative response to PD-1 blockade in mouse syngeneic tumor models[23]. In this study, we further explore the impact of pharmacological LSD1 inhibition on adoptive T-cell (ACT) therapy and find that targeting LSD1 by chemical inhibitors during the in vitro activation and expansion of T cells remodels the epigenome associated with a phenotype of increased effector cytokine production and decreased inhibitory receptor expression. Notably, a timely and transient inhibition of LSD1 during T-cell activation proves sufficient to alleviate T-cell exhaustion and improve the persistence and antitumor efficacy of adoptively transferred mouse OT1 cells. These effects of LSD1 inhibition are also consistently observed in human CD19-CAR T cells. Thus, our study reveals an exhaustion-promoting epigenetic program driven by LSD1 that could be targeted by pharmacological inhibitors to enhance the therapeutic effectiveness of adoptively transferred T cells in cancer treatment.

## Results

### Targeting LSD1 by chemical inhibitors increases effector cytokines and decreases inhibitory receptors in CD8+ T cells

To investigate whether LSD1 is a potential target for improving the functionality of CD8+ T cells in adoptive cell therapy, we first analyzed the impact of LSD1 inhibition on T-cell activation. We selected two well-known irreversible inhibitors, GSK2879552 (abbreviated as GSK) and ORY1001[24,25], for this purpose. In a cellular thermal shift assay (CETSA), both inhibitors increased the thermostability of LSD1 in CD8+ T cells (Supplementary Fig. 1a), confirming their on-target engagement. GSK and ORY1001 achieved complete target engagement in CD8+ T cells at

concentrations exceeding 167 nM and 2 nM, respectively (Supplementary Fig. 1b, c). We then treated CD8+ T cells, isolated from three separate OT1 mice and activated in vitro with anti-CD3/anti-CD28 for 48 h followed by IL-2 expansion for an additional 72 h, with 0.5 μM GSK. RNA-seq analysis of these samples showed that GSK treatment significantly altered the transcriptome, with 241 genes upregulated and 426 genes downregulated (Fig. 1a, b). Gene Ontology (GO) enrichment analysis revealed that the upregulated genes were highly enriched in biological processes related to cell killing and cytokine production (Fig. 1c). GSK-treated CD8+ T cells generally exhibited higher transcript levels of effector molecules and chemokines, but reduced expression of inhibitory receptors (Fig. 1d). Flow cytometry analysis confirmed that GSK treatment increased cytokine production of IL-2, TNFα, and IFNγ, while decreased cell surface expression of PD-1 and CD39, although GzmB and TIM-3 protein levels remained unchanged (Fig. 1e–g). Additionally, GSK elevated the expression of memory/progenitor-related genes, including SLAMF6 and CD62L (Fig. 1h, i). In support of the on-target effect of GSK on gene expression, we observed similar alterations in PD-1 and SLAMF6 expression in both *Lsd1* conditional knockout (*Cd4CreLsd1f/f*, cKO) and inducible knockout (*Rosa26Cre-ERT2Lsd1f/f*, iKO) CD8+ T cells as with GSK treatment (Supplementary Fig. 1d–i). Of note, GSK had minimal effects on PD-1 and SLAMF6 expression in the LSD1-deficient CD8+ T cells (Supplementary Fig. 1d–i). In addition, another LSD1 inhibitor, ORY1001, showed similar effects on gene expression as GSK (Supplementary Fig. 1j, k). These findings suggest that targeting LSD1 via chemical inhibitors could impact CD8+ T-cell function by modulating gene transcription.

Recent reports have highlighted several epigenetic modulators in adoptive T-cell therapy[15,16,20–22]. While GSK and ORY1001 showed no dose-limiting toxicities for CD8+ T cells in the examined range of 50–1000 nM, decitabine (a DNMT inhibitor) and chaetocin (a SUV39H1 inhibitor) caused severe cell death at concentrations exceeding 50 nM or 20 nM, respectively. At lower doses, decitabine had a dose-dependent effect of upregulating PD-1 but downregulating SLAMF6 and CD62L, contrasting the observations with chaetocin and LSD1 inhibitors (Supplementary Fig. 2a–d). GSK126, an EZH2 inhibitor, at 1 μM concentration appeared to enhance the effector phenotype of T cells, marked by increased expression of PD-1, CD44, and GzmB (Supplementary Fig. 2a–f). In contrast to other modulators, LSD1 inhibitors did not reduce Ki-67 expression in CD8+ T cells (Supplementary Fig. 2g), and modestly lowered apoptosis levels of activated CD8+ T cells (Supplementary Fig. 2h, i). This was linked to a higher proportion of T cells undergoing five or more rounds of cell division (Supplementary Fig. 3a), suggesting that LSD1 inhibitors outperform other epigenetic modulators by enhancing a memory/progenitor-associated phenotype and conferring a survival benefit to activated CD8+ T cells. However, GSK treatment did not noticeably affect the direct cytolytic activity of OT1 cells in vitro (Supplementary Fig. 3b–d).

### LSD1 inhibition reshapes the epigenome of CD8+ T cells

To investigate the regulatory mechanism of gene expression by LSD1 inhibition, we performed ChIP-seq assays with in vitro activated and expanded CD8+ T cells. LSD1 protein was expressed in unstimulated CD8+ T cells and notably increased upon TCR stimulation (Supplementary Fig. 4a). ChIP-seq analysis uncovered LSD1 binding peaks at many promoters (5,121/30,664) and defined enhancers (5,932/36,976) in Veh-treated CD8+ T cells (Fig. 2a, b). GSK treatment had little overall effect on LSD1 binding at promoters, but it deprived LSD1 from their binding sites at enhancers (Fig. 2a, b). Surprisingly, only a small subset of LSD1 binding sites, referred to as cluster 2 (C2, 517) of promoters and C2 (512) of enhancers, displayed the expected accumulation of H3K4me2 or H3K4me1 following GSK treatment (Fig. 2a, b), which are known as the direct substrates of LSD1 catalytic activity[26]. This indicates a potential scaffolding function of LSD1 independent of its histone demethylase activity, as has been reported elsewhere[27,28].

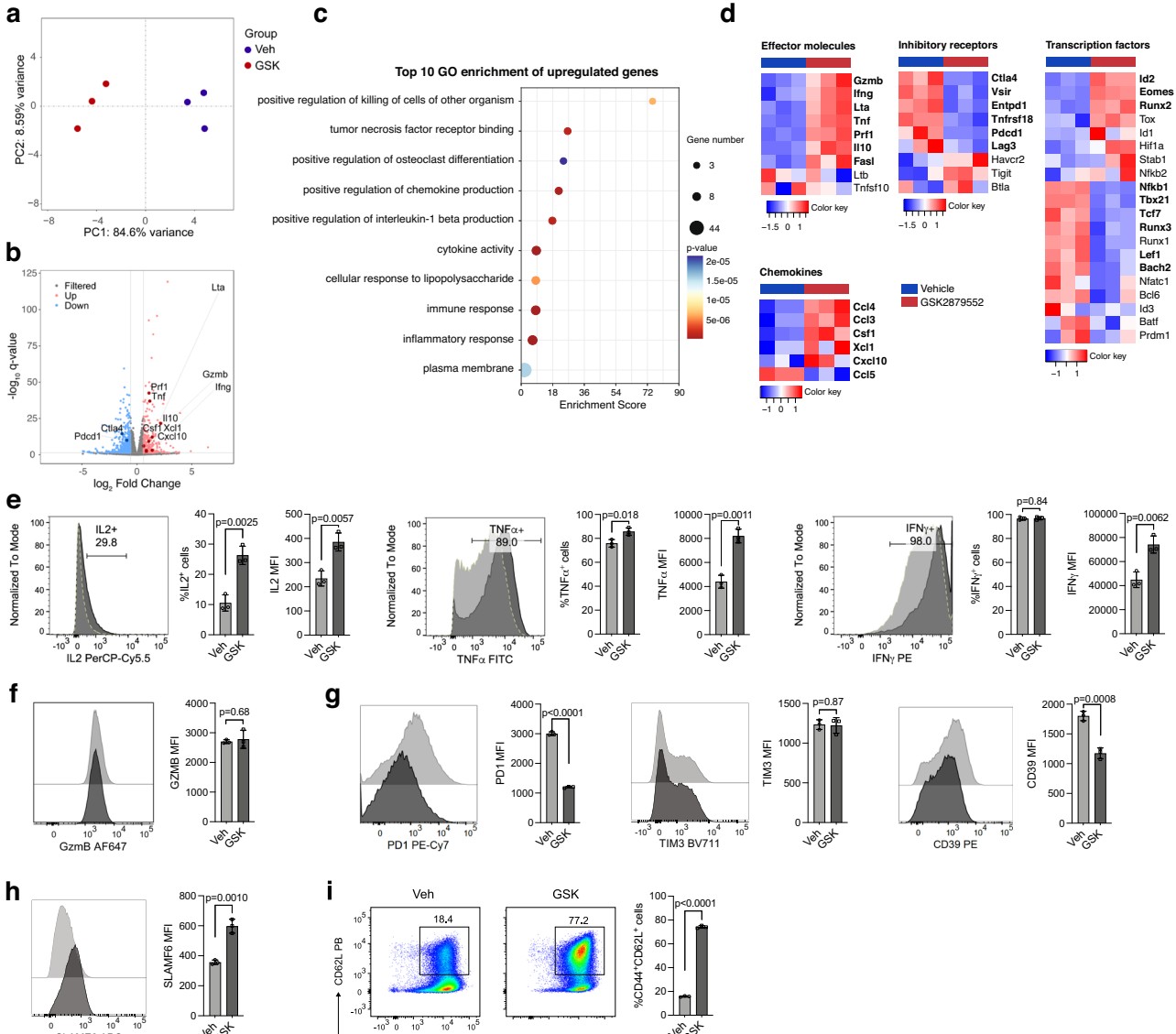

**Fig. 1 | LSD1 inhibition by GSK elevates cytokine production and decreases inhibitory receptor expression in in vitro activated CD8+ T cells. a** Principal component analysis (PCA) of RNA-seq data of mouse CD8+ T cells activated and expanded in vitro for 5 days with the treatment of GSK2879552 (GSK) or vehicle control (Veh) (*n* = 3 mice per group). **b** Volcano plot showing differential gene expression in RNA-seq analysis of GSK- versus Veh-treated CD8+ T cells (n = 3, [Fold change] > 1.5 and *q*-value < 0.05 as the cutoff]). **c** Dot map showing the top 10 terms in the GO analysis of the upregulated genes in the comparison of GSK- versus Veh-treated CD8+ T cells. *P*-values were calculated by one-sided Fisher's exact test and adjusted by the Benjamini-Hochberg method. **d** Heatmaps showing differential

gene expression of effector molecules, chemokines, inhibitory receptors, and transcription factors (*q* < 0.05, marked in bold). Flow cytometry of cytokines (**e**), GzmB (**f**), inhibitory receptors (**g**), and SLAMF6 (**h**) expressed in GSK- or Veh-treated CD8+ T cells (*n* = 3). MFI, mean fluorescence intensity. **i** Flow cytometry analysis of frequencies of CD44+CD62L+ cells in GSK- or Veh-treated CD8+ T cells (*n* = 3). Data in (**e**−**i**) are presented as mean ± standard deviation (SD) and are representative of three independent experiments. Statistical significance was determined by two-sided unpaired *t*-test (**e**−**i**). Source data are provided as a Source Data file.

At promoters, GSK treatment appeared to have a more robust effect on differential H3K27ac (Supplementary Fig. 4b). At a number of enhancers without LSD1 binding (C6 and C9), GSK treatment also resulted in H3K4me1 accumulation, which was associated with gained LSD1 binding peaks (Fig. 2c). Lack of apparent H3K27ac peaks at H3K4me1-enriched enhancers (C2, C6, and C9) indicated that most of them were probably not active in GSK-treated CD8+ T cells (Fig. 2b, c)[29].

By intersecting ChIP-seq and RNA-seq data, we found that a majority of upregulated effector and cytokine genes in GSK-treated CD8+ T cells showed no increases in Pol II occupancies at their promoters (Fig. 2d, the upper panel). In contrast, promoters of down-regulated inhibitory receptor (IR) genes mostly showed reduction in Pol II occupancies, correlating with the decreased H3K4me3 (Fig. 2d,

the lower panel). Inspection of individual gene tracks showed that active histone marks at the *Tnf* locus were not elevated in GSK-treated CD8+ T cells, despite an increase in RNA expression (Fig. 2e). The memory/progenitor-related gene *Slamf6* showed increased H3K4me3 and H3K27ac at its promoters upon GSK treatment, correlating with transcriptional activation (Fig. 2f). A putative enhancer was identified 50 kb downstream transcription start site (TSS) of *Slamf6*, which was marked by H3K4me1, H3K4me2 and H3K27ac and bound by LSD1 (Fig. 2f). GSK treatment led to an enrichment in both H3K4me1 and H3K4me2, the substrates of LSD1 catalytic activity, as well as H3K27ac at this enhancer (Fig. 2f), which may account for the upregulated *Slamf6* expression. EOMES, which is reported to be involved in the self-renewal of central memory T cells[30], also accumulated H3K4me1 and/

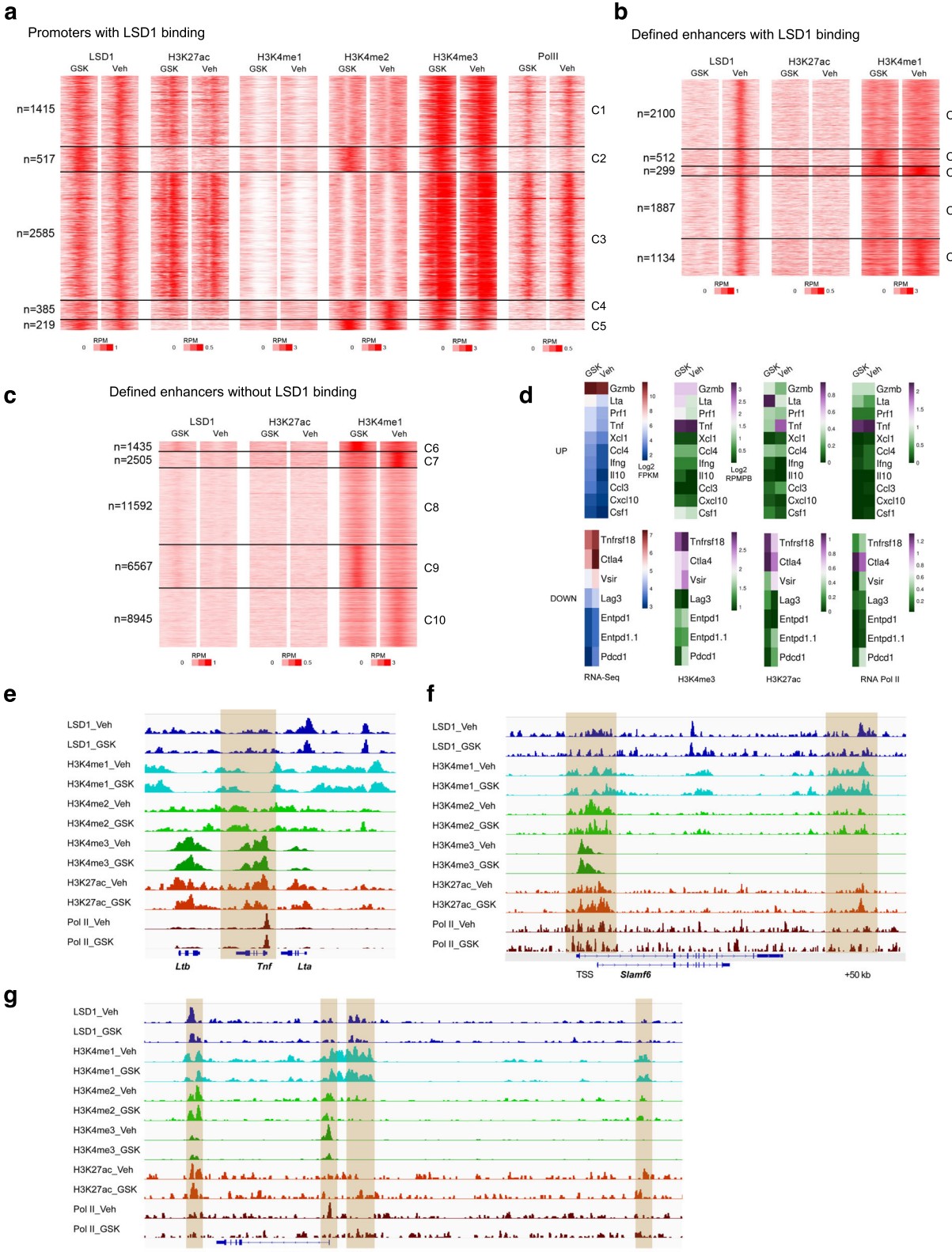

**Fig. 2 | GSK treatment reshapes chromatin landscapes of CD8⁺ T cells.**
**a** Heatmaps of ChIP-seq signals at LSD1-bound promoters (TSS ± 1 kb) in GSK- or Veh-treated CD8⁺ T cells, generated by K-means clustering on H3K4me2 signals. Heatmaps of ChIP-seq signals at defined enhancers (Peak ± 1 kb) with (**b**) or without (**c**) LSD1 binding, generated by K-means clustering on H3K4me1 signals.

**d** Heatmaps of ChIP-seq signals at promoters (TSS ± 1 kb) of upregulated effector and cytokine genes and downregulated inhibitory receptor genes. IGV snapshots showing ChIP-seq tracks at genomic loci of *Tnf* (**e**), *Slamf6* (**f**), and *Pdcd1* (**g**). Differential peaks were determined by $p < 0.05$, with changes over 20% up/down. The presented data are from one of two repeated experiments (**a**–**g**).

or H3K4me2 at putative cis-regulatory elements upon LSD1 inhibition (Supplementary Fig. 4c). GSK treatment decreased Pol II occupancies at the promoter of *Pdcd1* (encoding PD-1), accompanied by a significant reduction in the active histone mark H3K4me3, while other histone marks showed minimal changes at either the promoter or previously reported enhancers (Fig. 2g)[31]. A similar pattern was observed for *Entpd1* that encodes CD39 (Supplementary Fig. 4d). These results indicate that LSD1 inhibition impacts functionally different sets of genes with distinct epigenetic mechanisms.

### LSD1 responds to TCR activation and IL-2 signaling to promote PD-1 expression

We further investigated the regulation of IR expression by LSD1. PD-1 was increasingly expressed by raising the strength of TCR stimulation (Fig. 3a). Without TCR activation, CD8+ T cells expressed a low basal level of PD-1, which was not much affected by GSK treatment (Fig. 3b). When GSK treatment was only applied during the IL-2 expansion phase post-TCR stimulation (from day 4 to 8), it also lost the suppressive effect on IR expression including PD-1, TIM-3, and CD39, although it still can upregulate SLAMF6 expression (Fig. 3c). To further support this, we showed that depleting LSD1 post-TCR activation in the *Rosa26^Cre-ERT2^Lsd1^f/f^* CD8+ T cells, using 4-hydroxytamoxifen (4-OHT) during the IL-2 expansion phase (4OHT-E), failed to effectively inhibit PD-1 and TIM-3 expression (Fig. 3d, e). Yet, it retained the capacity to increase cytokine production and CD62L expression (Supplementary Fig. 5a, b). These results underscore the importance of simultaneous inhibition of LSD1 during TCR stimulation, implicating that GSK treatment suppresses IR expression possibly through interfering with the TCR signaling pathway. However, comparable calcium influx and CD44 expression were detected in GSK- and Veh-treated CD8+ T cells (Fig. 3f, g), which indicates that GSK treatment unlikely affects the TCR signaling transduction. Instead, it might directly impede the transcription factor (TF)-mediated IR transcription in response to TCR activation, warranting further investigation.

IL-2, supplemented to support in vitro T-cell expansion, also upregulated PD-1 expression in a dose-dependent manner (Fig. 3h), as reported elsewhere[32]. Blocking the IL-2 signaling pathway with IL-2-neutralizing antibodies or a JAK inhibitor (tofacitinib) significantly decreased PD-1 expression (Fig. 3i, j). Under these conditions, GSK treatment no longer had a suppressive effect on PD-1 expression, despite retaining the inducible effect on SLAMF6 expression (Fig. 3k). Similarly, in the *Cd4^Cre^Stat5^f/f^* (*Stat5^cKO^*) CD8+ T cells, the suppressive effect of GSK treatment on PD-1 expression was by and large diminished (Fig. 3l). These results suggest that LSD1 inhibition suppresses PD-1 expression in a way depending on the activity of the IL-2/STAT5 pathway. Since LSD1 is a nuclear histone modifier, it may not be surprising that the main signaling pathways downstream of the IL-2 receptor were not affected by GSK treatment (Fig. 3m). We speculated that LSD1 may regulate the expression of IL-2 target genes, which in turn affected PD-1 expression, considering that LSD1-mediated histone demethylation was not observed at cis-regulatory elements of the *Pdcd1* locus (Fig. 2g). *Eomes*, a target gene of IL-2 signaling[33], was found to be directly targeted by LSD1 and upregulated by GSK treatment (Fig. 1d, Supplementary Fig. 4c), as confirmed by flow cytometry (Fig. 3n). Depletion of EOMES by CRISPR-mediated knockout significantly increased PD-1 expression, whereby the suppressive effect of GSK treatment was largely compromised (Fig. 3o, Supplementary Fig. 5c). Thus, LSD1 inhibition by GSK likely facilitates EOMES expression induced by IL-2 stimulation to counteract IL-2-induced PD-1 expression. In line with the role of EOMES in the self-renewal of central memory T cells[30], EOMES depletion reduced the percentage of CD44+CD62L+ population elevated by GSK treatment (Supplementary Fig. 5d). Our results therefore implicate a mode that LSD1 inhibition suppresses PD-1 expression possibly through elevating negative regulators like EOMES.

### LSD1 inhibition attenuates the induction of T-cell exhaustion by persistent TCR activation in vitro

The observed suppressive effect of LSD1 inhibition on IR expression prompted us to examine its impact on T-cell exhaustion. To induce exhaustion, in vitro activated CD8+ T cells (Teff) were repeatedly stimulated with plate-bound anti-CD3 with the addition of IL-2 every other day across four rounds (Fig. 4a). The resulting cells exhibited markedly higher levels of PD-1, TIM-3, and CD39 compared with Teff cells expanded with only IL-2 (Fig. 4b–d), with most being triple-positive for these markers (Fig. 4e), indicating a state of terminal exhaustion. GSK treatment throughout exhaustion induction significantly attenuated the upregulation of PD-1 and CD39 and subsequently decreased the frequencies of triple-positive cells (Fig. 4b–e). Corresponding to high-level expression of IRs, exhausted CD8+ T cells (Tex) showed impaired TNFα production, a deficit substantially mitigated by GSK treatment (Fig. 4f). GSK treatment also elevated IFNγ production in Tex cells (Fig. 4f). We further assessed the cytolytic activity and found that, compared with Teff cells (Supplementary Fig. 3b), exhausted OT1 (OT1ex) cells after multiple rounds of stimulation were impaired in killing target B16-OVA cells (Fig. 4g). In consistence with the less exhausted phenotype, GSK-treated OT1ex cells displayed an improved antigen-specific killing effect (Fig. 4g), in contrast to the earlier demonstration that GSK-treated Teff cells showed no cytolytic advantage over control Teff cells (Supplementary Fig. 3d). We also observed that the CD62L+ population, reported to represent the progenitor of Tex cells[34], disappeared following repeated TCR stimulation, which was partially reversed by GSK treatment (Fig. 4h). The suppressive effect of LSD1 inhibitors on T-cell exhaustion phenotype was recapitulated by genetic ablation of *Lsd1* (Supplementary Fig. 6a–e).

To further corroborate the above findings, we utilized a co-culture system, where activated OT1 cells were repeatedly stimulated with fresh tumor cells expressing the OVA antigen (Fig. 4i). After four-day co-culture, most PD-1+ OT1 cells co-expressed TIM-3 and CD39 (Fig. 4j), indicating exhaustion. As expected, GSK treatment reduced the expression of TIM-3 and CD39 by OT1ex cells, leading to a lower frequency of TIM-3+CD39+ population (Fig. 4j). In addition, GSK treatment increased the production of TNFα and IL-2 by OT1ex cells, despite the unaltered IFNγ production (Fig. 4k). The frequency of CD62L+ population within OT1ex cells was also markedly elevated by GSK treatment (Fig. 4l). Thus, in both models of in vitro exhaustion, LSD1 inhibition by GSK treatment consistently alleviated T-cell exhaustion.

### A timely and transient inhibition of LSD1 during T-cell activation suffices to augment the antitumor potency of adoptively transferred T cells

We next sought to integrate GSK treatment into adoptive transfer modalities. OVA-expressing B16 melanoma tumors were established by subcutaneous injection into wild-type mice for 8 to 10 days, followed by the transfer of OT1 cells that were activated and expanded in vitro with or without GSK treatment for 5 days (referred to as GSK priming). No additional GSK treatment was performed post transfer. Transfer of 1 million control OT1 cells showed negligible effects on B16-OVA tumor growth, but the administration of an equivalent number of GSK-primed OT1 cells significantly inhibited tumor growth and prolonged animal survival (Fig. 5a, b). The enhanced antitumor effect, as also observed with ORY1001-primed OT1 cells, was consistently seen in the Yumm1.7-OVA tumor model (Fig. 5c, d). Of note, the potency of GSK priming in amplifying the antitumor efficacy of OT1 cells was on par with that of chaetocin priming but stronger than that of decitabine and GSK126 priming (Supplementary Fig. 7a, b), which was correlated with their individual effects on T-cell activation (Supplementary Fig. 2a–h). Over two weeks after the transfer, mice receiving either GSK-primed or

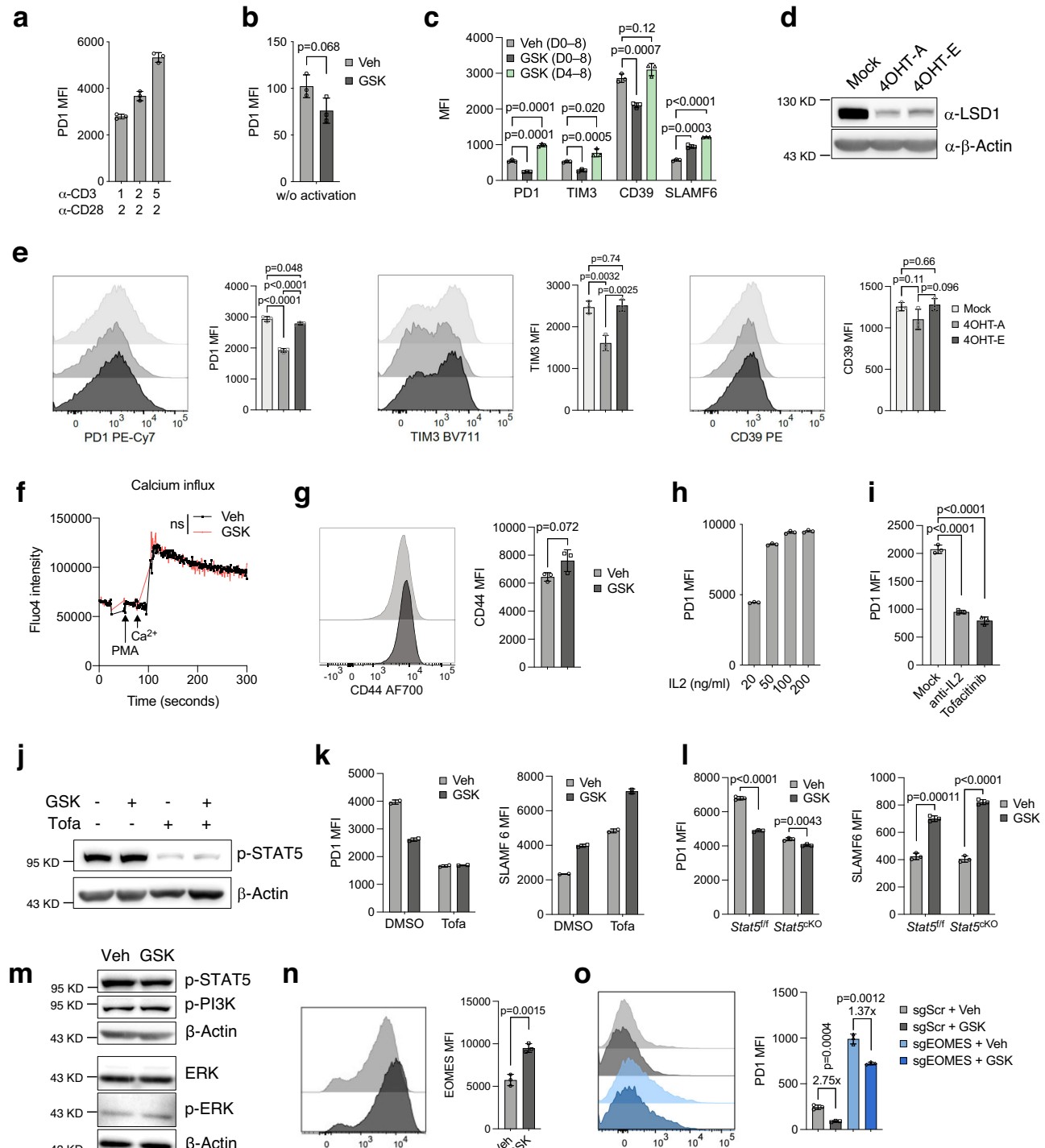

**Fig. 3 | LSD1 responds to both TCR and IL-2 signaling to induce IR expression.** **a** Flow cytometry analysis of PD-1 expression in CD8⁺ T cells activated with different concentrations (µg/ml) of anti-CD3/anti-CD28 for 2 days and expanded with IL-2 for 3 days ($n = 3$). **b** Flow cytometry analysis of PD-1 expression in unstimulated CD8⁺ T cells cultured with IL-2 and IL7 for 5 days in the presence or absence of GSK ($n = 3$). **c** Flow cytometry analysis of inhibitory receptors (IRs) and SLAMF6 in CD8⁺ T cells after 2-day activation with anti-CD3/anti-CD28 and 6-day expansion with IL-2, during which GSK was added as indicated ($n = 3$). Immunoblot of LSD1 (**d**) and flow cytometry analysis of IR expression (**e**, $n = 3$) in *Rosa26^{Cre-ERT2}Lsd1^{f/f}* CD8⁺ T cells on day 5, treated with 4-OHT during the 2-day TCR activation period (4OHT-A) or the 3-day IL-2 expansion period (4OHT-E). **f** Flow cytometry analysis of Ca²⁺ influx in GSK- or Veh-treated CD8⁺ T cells in response to PMA stimulation ($n = 3$). ns, not statistically significant. **g** Flow cytometry of CD44 in GSK- or Veh-treated CD8⁺ T cells on day 5 ($n = 3$). **h** Flow cytometry of PD-1 in CD8⁺ T cells after 2-day

activation and 3-day expansion with different concentrations of IL-2 ($n = 3$). **i** Flow cytometry of PD-1 in CD8⁺ T cells treated with anti-IL-2, tofacitinib, or vehicle control (Mock) ($n = 3$). **j** Immunoblot of p-STAT5 in CD8⁺ T cells treated with GSK and/or tofacitinib (Tofa). **k, l** Flow cytometry of PD-1 and SLAMF6 in wild-type (**k**, $n = 2$), *Stat5^{f/f}*, and *Cd4^{Cre}Stat5^{f/f}* (*Stat5^{cKO}*) (**l**, $n = 3$) CD8⁺ T cells with the indicated treatments for 5 days. **m** Immunoblots of IL-2 signaling proteins in CD8⁺ T cells treated with GSK or Veh for 5 days. **n** Flow cytometry of EOMES in GSK or Veh-treated CD8⁺ T cells ($n = 3$). **o** Flow cytometry of PD-1 in Rosa26-Cas9 CD8⁺ T cells transduced with sgScramble or sgEOMES and treated with GSK or Veh for 5 days ($n = 3$). Data in this figure are presented as mean ± SD and are representative of three (**a–g, n, o**) or two (**h–m**) independent experiments. Statistical significance in this figure was determined by two-sided unpaired *t*-test. Source data are provided as a Source Data file.

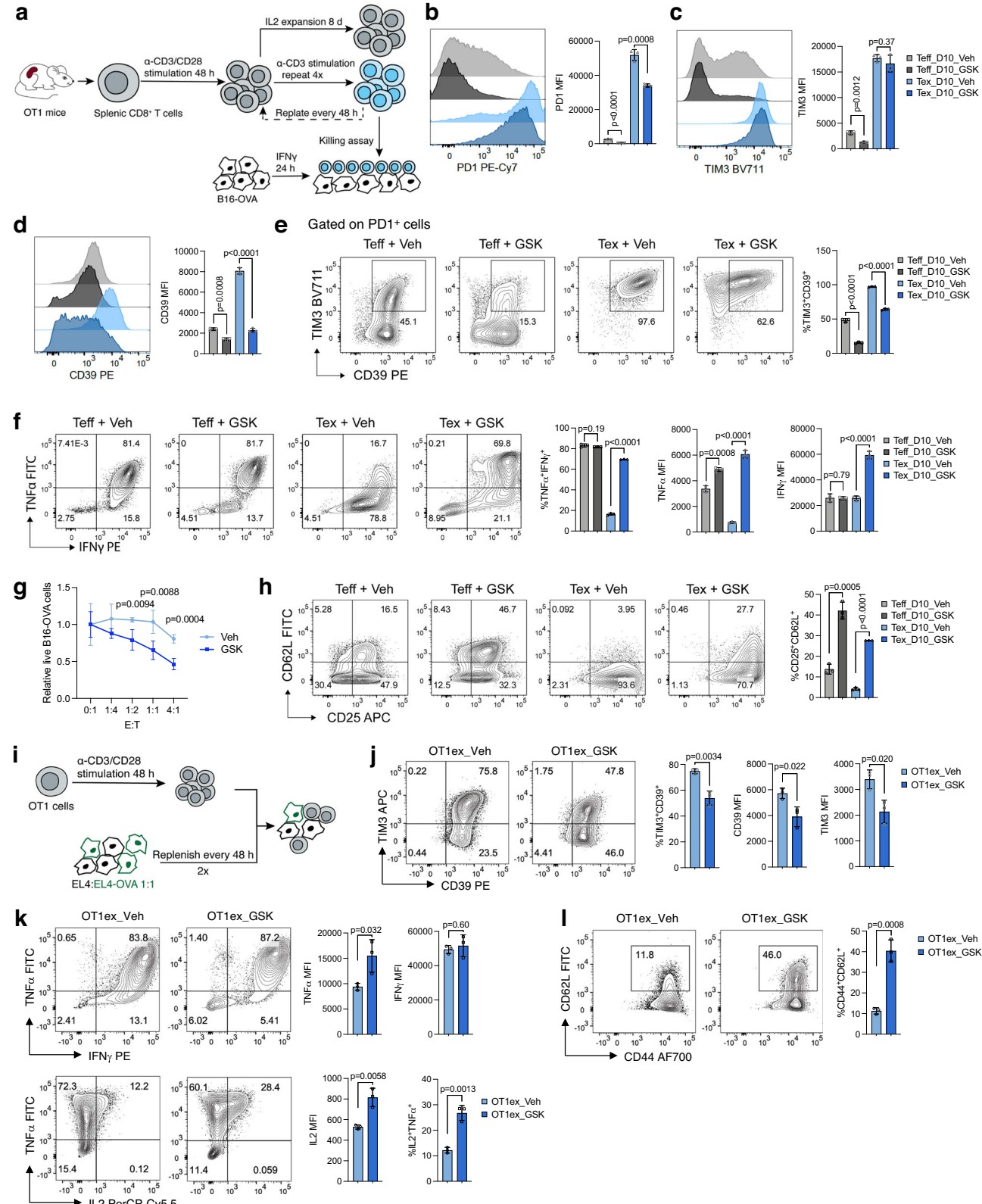

control OT1 cells did not show apparent body weight loss (Supplementary Fig. 7c). No increase in systemic levels of IL-6 or IL-1β, two cytokines linked to cytokine release syndrome and neurotoxicity[35,36], was detected in any group of recipient mice (Supplementary Fig. 7d, e). Further, no evident lesions or immune cell infiltration were noted in the histological sections of the brain, heart, or liver (Supplementary

Fig. 7f). These data suggest that GSK priming enhances the antitumor efficacy of adoptively transferred T cells without leading to toxicities.

To determine whether the brief period of GSK treatment during OT1 cell in vitro activation had a lasting impact on T-cell exhaustion in the TME, we analyzed the tumor-infiltrating lymphocytes (TILs) on day 5 post transfer. We found that GSK-primed OT1 cells were presented in

**Fig. 4 | LSD1 inhibition impedes the in vitro induction of T-cell exhaustion.**
**a** Experimental design for in vitro induction of exhaustion by repeated anti-CD3
stimulation and functional assessment of exhausted CD8⁺ T cells. Flow cytometry of
PD-1 (**b**), TIM-3 (**c**), and CD39 (**d**) in CD8⁺ T cells after repeated stimulation (Tex) or
continuous IL-2 expansion (Teff) for 10 days with or without GSK treatment (n = 3).
**e** Percentages of TIM-3⁺CD39⁺ cells among PD-1⁺ Teff and Tex cells (n = 3). **f** Flow
cytometry analysis of cytokine production in GSK- or Veh-treated Teff and Tex cells
(n = 3). **g** Cytotoxicity of GSK- or Veh-treated OT1 cells after repeated stimulation,
assessed by antigen-dependent killing assay (n = 4). E:T, effector to target ratio.

**h** Percentages of CD25⁺CD62L⁺ cells among GSK- or Veh-treated Teff and Tex cells
(n = 3). **i** Experimental design for in vitro induction of exhaustion in OT1 cells by co-
culture. Flow cytometry analysis of IR (**j**) and cytokine (**k**) expression in exhausted
OT1 (OT1ex) cells with or without GSK treatment (n = 3). **l** Percentages of
CD44⁺CD62L⁺ cells among GSK- or Veh-treated OT1ex cells (n = 3). Data in this
figure are presented as mean ± SD and are representative of three (**b**–**f**, **h**) or two
(**g**, **j**–**l**) independent experiments. Statistical significance in this figure was deter-
mined by two-sided unpaired t-test. Source data are provided as a Source Data file.

the TME at a considerably higher frequency compared with Veh con-
trol (Fig. 5e), indicating better persistence. Further, GSK-primed OT1
TILs displayed reduced levels of PD-1 and CD39, despite similar TIM-3
levels (Fig. 5f–h), and contained a higher frequency of the CD62L⁺
population (Fig. 5i). These phenotypic changes were in line with pre-
vious observations that GSK treatment attenuated the in vitro induc-
tion of T-cell exhaustion, suggesting that transient GSK priming can
mitigate exhaustion in adoptively transferred T cells and enhance their
antitumor potency.

Considering the lasting effect of LSD1 inhibition on PD-1 expres-
sion, we investigated whether this could primarily account for the
enhanced antitumor potency of GSK-primed OT1 cells. To this end, we
utilized PD-1-blocking antibodies to normalize the effect of differential
PD-1 expression on the adoptively transferred OT1 cells. Under these
conditions, GSK-primed OT1 cells continued to exhibit a stronger
antitumor effect against B16-OVA tumors compared with Veh control
(Fig. 5j, k). This suggests that GSK priming operates through
mechanisms beyond merely reducing PD-1 levels. Further, GSK-primed
OT1 cells cooperated with PD-1 blockade to inhibit B16-OVA tumor
growth, a synergy not observed with control OT-1 cells (Fig. 5j, k).

Our in vitro observation points out that the period of TCR sti-
mulation is crucial for the effective implementation of LSD1 inhibition
in suppressing IR expression. To explore the biological relevance of
this finding, we took advantage of the inducible knockout of LSD1 in
the isolated Rosa26^{Cre-ERT2}Lsd1^{f/f} OT1 cells. Depletion of LSD1 by 4-OHT
treatment during the TCR activation phase (day 0–2) enabled the
adoptively transferred OT1 cells to effectively control B16-OVA tumor
growth (Fig. 5l). In stark contrast, OT1 cells undergoing LSD1 depletion
during the IL-2 expansion phase post-TCR stimulation (4-OHT treat-
ment during day 2–4) did not impact tumor growth, similar to cells
retaining LSD1 function (no 4-OHT treatment) (Fig. 5l). Thus, our data
underscore that LSD1 inhibition should occur simultaneously with TCR
stimulation to accomplish therapeutic enhancement of adoptive T-cell
therapy.

To assess the impact of GSK priming duration on the antitumor
potency, we administered GSK to in vitro activated OT1 cells for 2, 4,
and 6 days, respectively, before transferring them to tumor-bearing
mice after 6 days of total culture. Remarkably, 2-day GSK priming
during TCR stimulation greatly improved the antitumor efficacy of
OT1 cells compared with vehicle-treated counterparts (Fig. 5m).
Extending GSK priming to 6 days yielded incremental but less
marked enhancements in efficacy (Fig. 5m). However, the magnitude
of change in gene expression (e.g., PD-1, SLAMF6, and CD62L)
induced by 6-day GSK priming was much higher than that by 2-day
priming (Fig. 5n). This indicates that the antitumor efficacy of GSK
priming does not directly correlate with the fold changes in gene
expression. Using the CETSA assay, we confirmed that the initial
addition of 0.5 μM GSK completely inhibited LSD1 during the 2-day
TCR stimulation period; removing GSK subsequently during the IL-2
expansion phase released the inhibition (Fig. 5o, p). Therefore, 2-day
GSK priming concurrent with TCR stimulation is the key parameter
and, to a large extent, sufficient for LSD1 inhibition to potentiate the
antitumor efficacy of OT1 cells.

## LSD1 inhibitor-primed CD8⁺ T cells demonstrate both enhanced antigen-dependent and -independent persistence after adoptive transfer

Limited persistence of adoptively transferred T cells in vivo, in part due
to exhaustion, is a major hurdle to their therapeutic effectiveness in
cancer treatment[37,38]. We next asked whether GSK-primed OT1 cells
had better persistence in the TME, as these cells exhibited less
exhaustion. By analyzing OT1 cells retrieved from B16-OVA tumors, we
found that GSK-primed OT1 TILs displayed a similar proliferation rate
to control cells (Fig. 6a), but became more resistant to apoptosis,
indicated by lower Annexin V staining (Fig. 6b, c). This survival
advantage likely contributed to their higher frequency in the TME, as
previously observed (Fig. 5e). To assess whether this acquired resis-
tance to apoptosis was a cell-intrinsic property or influenced by the
TME, we co-transferred GSK-primed (CD45.2) and Veh-primed
(CD45.1/CD45.2) OT1 cells into the same recipient mice (CD45.1) car-
rying B16-OVA tumors and then analyzed the TILs (Fig. 6d). On day 5
after co-transfer, GSK-primed OT1 cells were more abundant than
control OT1 cells in the TME (Fig. 6e), which was associated with
reduced levels of Annexin V staining (Fig. 6f), thus demonstrating their
inherent resistance to apoptosis due to GSK priming. Meanwhile, PD-1
expression was significantly lower in GSK-primed OT1 TILs compared
with control OT1 cells (Fig. 6g). Importantly, GSK-primed OT1 cells
maintained their intratumoral abundance on day 8 after co-transfer,
while in contrast control OT1 TILs experienced a ~40% reduction in
frequency from day 5 (Fig. 6e). Thus, GSK priming appears to enhance
intratumoral persistence of adoptively transferred T cells, possibly
because of a survival advantage.

To determine whether antigen stimulation was required for this
improved intratumoral persistence, we transferred GSK- and Veh-
primed OT1 cells into B16 tumor-carrying mice respectively, and ana-
lyzed TILs on day 5. In the absence of cognate OVA antigen, OT1 cells
were rarely detected in the TME regardless of GSK priming, along
which similar tumor sizes were observed (Fig. 6h, i). This implies that
tumor antigen stimulation drives intratumoral expansion of adoptively
transferred T cells, on which GSK priming imposes a promoting effect.
Despite in low abundance, GSK-primed OT1 TILs tended to have a
higher frequency than control cells, albeit not statistically significant
(Fig. 6i). Consistently, GSK priming raised the frequencies of trans-
ferred OT1 cells in the tumor-draining lymph nodes (TdLNs), albeit at
low abundance (Fig. 6j), together suggesting that GSK priming could
also promote antigen-independent persistence of transferred T cells.
To support this notion, we co-transferred of GSK- and Veh-primed
OT1 cells into naïve mice, followed by OVA₂₅₇₋₂₆₄ immunization. Four
days post-immunization, we observed significantly higher frequencies
of GSK-primed OT1 cells than control cells in the spleens and increased
frequencies of IL7Rα⁻KLRG1⁺ effector T cells derived from GSK-primed
OT1 cells (Fig. 6k, l). Tracking up to 28 days post-immunization, we
confirmed that GSK-primed OT1 cells consistently persisted in higher
frequencies in the peripheral blood of recipient mice compared with
control cells (Fig. 6m). Together, these data support that LSD1 inhibi-
tion during T-cell activation enhances both antigen-dependent and
-independent persistence following adoptive transfer.

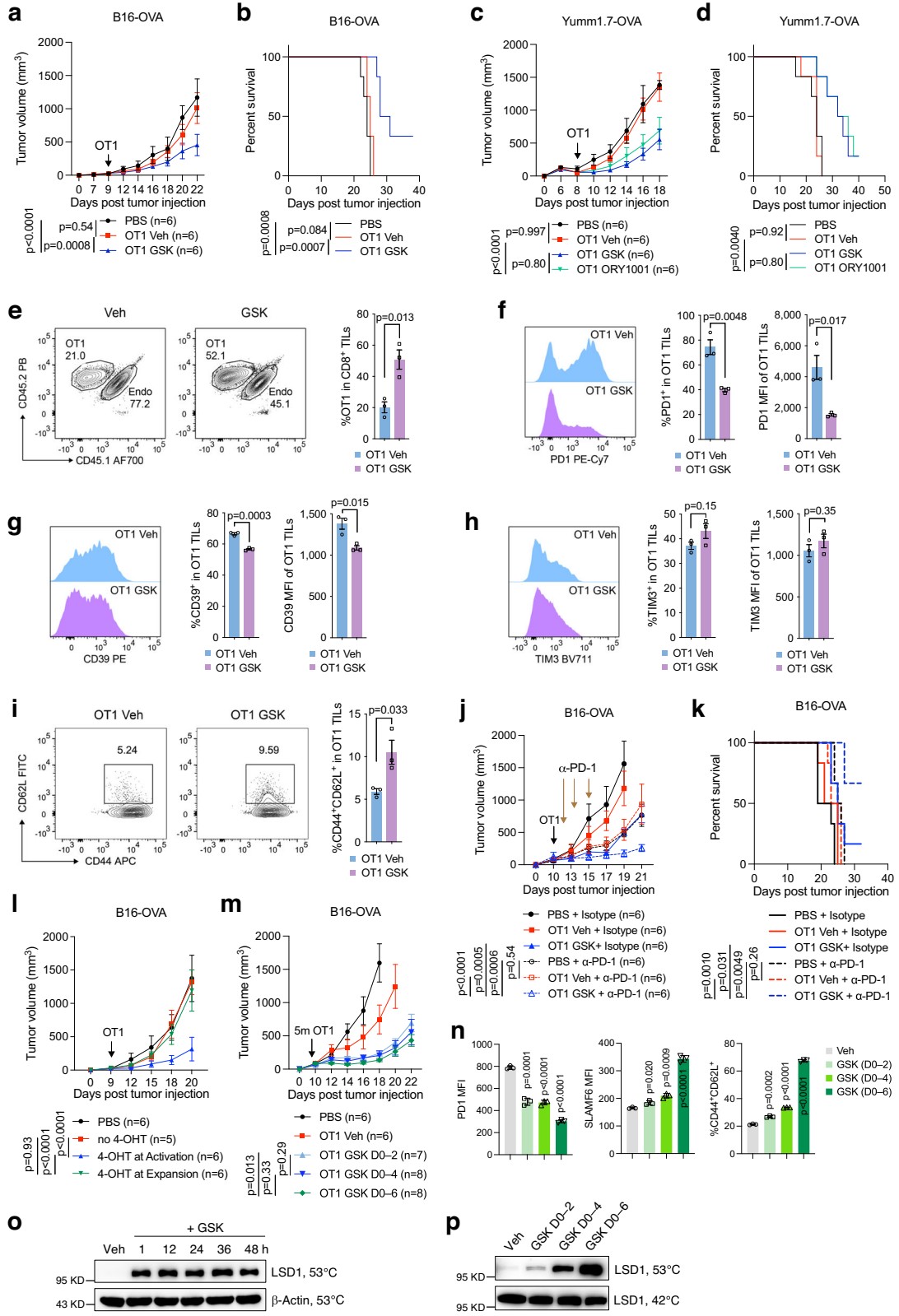

## Human CD19-CAR T cells primed with LSD1 inhibitors exhibit better persistence and superior antitumor efficacy

To evaluate the impact of pharmacological LSD1 inhibition on human T cells, we isolated peripheral blood mononuclear cells (PBMCs) from donors. T cells from PBMCs were stimulated with anti-CD3/CD28 beads for 2 days in the presence of IL-2, followed by an 8-day expansion with IL-2. In this process, GSK treatment significantly increased IL-2 and

IFNγ production by CD8[+] T cells (Fig. 7a), paralleling our findings in mice (Fig. 1e), although it did not obviously affect TNFα expression. Further analysis revealed that GSK treatment increased the proportion of polyfunctional CD8[+] T cells capable of producing three cytokines (Fig. 7b). In contrast to the observations in mice (Fig. 1f), GSK-treated human CD8[+] T cells displayed an increased expression of GzmB and a reduced frequency of Ki-67 positive cells (Fig. 7c). We also probed PD-1

**Fig. 5 | LSD1 inhibitor-primed OT1 cells show reduced exhaustion and enhanced antitumor efficacy.** Tumor growth and survival curves of mice subcutaneously inoculated with B16-OVA (**a**, **b**) or Yumm1.7-OVA (**c**, **d**) tumor cells and transferred with 1 million GSK-, ORY1001- or Veh-primed OT1 cells, or PBS as indicated (*n* = 6 mice per group). **e** Abundance of transferred CD45.2 OT1 cells in B16-OVA tumors, detected by flow cytometry on day 5 post transfer (*n* = 3 mice per group). Flow cytometry analysis of PD-1 (**f**), CD39 (**g**), and TIM-3 (**h**) expression in B16-OVA tumor-infiltrating OT1 cells (TILs) on day 5 post transfer (*n* = 3 mice per group). **i** Percentages of CD44$^+$CD62L$^+$ cells in OT1 TILs (*n* = 3 mice per group). Tumor growth (**j**) and survival curves (**k**) of B16-OVA tumor-bearing mice transferred with 1 million GSK- or Veh-primed OT1 cells or PBS, and injected with anti-PD1 or isotype control as indicated (*n* = 6 mice per group). **l** Tumor growth curves of B16-OVA tumor-bearing mice transferred with PBS or *Rosa26*$^{Cre-ERT2}$*Lsd1*$^{f/f}$ OT1 cells with vehicle or 4-OHT treatment during the 2-day activation period (4-OHT at

Activation) or the 3-day expansion period (4-OHT at Expansion) (*n* = 5–6 mice per group as indicated). **m** Tumor growth curves of B16-OVA tumor-bearing mice transferred with 5 million OT1 cells primed with GSK for different time durations (*n* = 6–8 mice per group as indicated). **n** Flow cytometry of PD-1, SLAMF6, and CD62L in OT1 cells treated with GSK for the indicated times during in vitro activation and expansion and analyzed on day 6 (*n* = 3). **o** CETSA assay detecting soluble LSD1 in activated CD8$^+$ T cells collected after treatment with 0.5 µM GSK for the indicated hours and heated at 53 °C. **p** CETSA assay detecting soluble LSD1 in activated CD8$^+$ T cells treated with 0.5 µM GSK for the indicated time durations and collected after 6 days of total culture. Data are representative of three (**a**, **b**) or two (**c–l**, **n–p**) independent experiments and are presented as mean ± SEM (**a–m**) or mean ± SD (**n**). Statistical significance was determined by two-sided unpaired *t*-test (**e–i**, **n**), two-way ANOVA (**a**, **c**, **j**, **l**, **m**), or log-rank test (**b**, **d**, **k**). Source data are provided as a Source Data file.

and TIM-3 expression by those expanded human CD8$^+$ T cells, but their low protein levels precluded the assessment of GSK's effect.

We then proceeded to generate CAR T cells to evaluate functional changes in response to GSK treatment. Human T cells activated in vitro were transduced with a lentiviral plasmid encoding CD19-specific CAR and fluorescent EGFP, which resulted in detectable CAR expression on the surface of T cells (Fig. 7d). Continuous 10-day GSK treatment did not appear to alter CAR expression or affect the cytolytic effect of CD19-CAR T cells against CD19-positive Nalm6 cells (Fig. 7d, e). However, when CD19-CAR T cells were exposed to multiple rounds of anti-CD3 stimulation, their cytolytic effect against Nalm6 and Raji cells was severely dampened, whereby GSK treatment displayed a rescue effect (Fig. 7f, g).

To assess the impact of GSK priming on CAR T-cell antitumor efficacy, we treated NCG mice bearing the Nalm6-lucif-EGFP leukemia with a low dose (2 × 10$^5$) of GSK-primed or control CD19-CAR T cells, or mock T cells without CAR transduction. Control CD19-CAR T cells and mock T cells did not exhibit any therapeutic effect, while GSK-primed CD19-CAR T cells strongly inhibited leukemia progression and extended animal survival (Fig. 7h–j). Corresponding to their antitumor efficacy, control CD19-CAR CD8$^+$ T cells were detected at a very low number in the peripheral blood on day 12 post-infusion, whereas the abundance of GSK-primed CD19-CAR T cells was markedly elevated (Fig. 7k–m). In a human melanoma model A375 (stably transduced with CD19), GSK priming also significantly enhanced the antitumor efficacy of CD19-CAR T cells, leading to tumor regression (Fig. 7n). In both models, the infusion of either control or GSK-primed CD19-CAR T cells did not cause severe body weight loss, though Nalm6-bearing mice lost weight at a later stage due to disease progression (Supplementary Fig. 7g, h). Thus, priming with LSD1 chemical inhibitors enhances the persistence and antitumor efficacy of human CAR T cells.

## Discussion

The efficacy of adoptively transferred CAR T and TCR T cells in cancer treatment is often limited by a variety of obstacles, including poor persistence in part due to exhaustion. In this study, we have shown that the transient and timely inhibition of LSD1 using clinically tested inhibitors during the in vitro activation and expansion of T cells reduces exhaustion and improves persistence, likely through rewiring chromatin landscapes, which consequently potentiates the antitumor efficacy of adoptively transferred T cells in both leukemia and solid tumor models.

Recent studies have uncovered epigenetic modifiers, including DNMTs and SUV39H1, involved in establishing the exhaustion-associated chromatin landscapes, presumably through installing the repressive chromatin methylation, specifically methylation on CpG of genomic DNA (5mC) and K9 of histone 3 (H3K9me3), which silence self-renewal and survival genes in CAR T cells[15,16,21,22]. EZH2, which also catalyzes repressive methylation, H3K27me3, nevertheless has been suggested to facilitate the functional reinvigoration

of CAR T cells[20]. Considering the dynamic and reversible nature of methylation, chromatin demethylases are also anticipated to play an important part in regulating CAR T-cell exhaustion. TET2, responsible for removing 5mC on DNA, has been studied in CAR T cells[18,19], but thus far histone demethylases are underexplored in this context. Our study somehow fills this gap by demonstrating the role of LSD1. Our ChIP-seq results suggest that LSD1 regulates functionally distinct gene sets by reshaping different histone marks, such as removing H3K4me1 and H3K4me2 at enhancers of memory/progenitor-related genes directly or indirectly facilitating the accumulation H3K4me3 at promoters of IR genes. The observation that histone demethylation by LSD1 only occurred at a small subset of LSD1-bound promoters and enhancers is consistent with previous reports[27,28], highlighting a potential scaffold function of LSD1 in gene regulation. The interaction with TFs, such as Oct4[39], has been proposed to inhibit the histone demethylation activity of LSD1, which might be an explanation for the unchanged H3K4me1 and H3K4me2 at LSD1 binding sites. The genome-wide alteration of H3K4me3 and H3K27ac during T-cell exhaustion has also been recently reported[10]. The fact that diverse chromatin modifications are involved in T-cell exhaustion indicates that perhaps co-targeting multiple chromatin modifiers is a more effective way to prevent T-cell exhaustion or even to revert exhausted T cells. A comprehensive profiling of chromatin landscapes, including the genome-wide distributions of key histone modifiers, histone modifications, and reader proteins, will provide new insights into the regulatory mechanisms of T-cell exhaustion, going beyond what ATAC-seq alone can offer.

The concurrence of LSD1 inhibition and TCR stimulation appeared to be crucial for the suppressive effect of LSD1 inhibitors on IR expression in CD8$^+$ T cells. Since the TCR signaling was unlikely affected by LSD1 inhibitors, reasonable speculation is that LSD1 may interact with TFs activated by TCR stimulation and cooperatively reshape the chromatin landscapes that derepress IR genes, but experimental evidence is warranted to test this hypothesis. This chromatin reshaping appeared to have a lasting effect, as the GSK-primed OT1 cells exhibited less exhaustion and better persistence after adoptive transfer in the absence of additional intervention. Post-TCR stimulation, the administration of LSD1 inhibitors lost the suppressive effect on IR expression, which highlights a particular time window for targeting LSD1, consistent with the speculation of an interplay between LSD1 and TCR signaling-activated TFs. Correspondingly, the concurence of LSD1 inhibition with TCR stimulation was a key factor for improving the antitumor efficacy of adoptively transferred T cells, which could not be compensated by a later perturbation of LSD1. Remarkably, the full potential of LSD1 inhibition to enhance T-cell antitumor efficacy was largely achieved with as little as 2-day exposure to pharmacological inhibitors, which was done concurrently with TCR stimulation. Thus, our study not only demonstrates LSD1 as a promising target for improving ACT therapy but also raises a possible interventional approach.

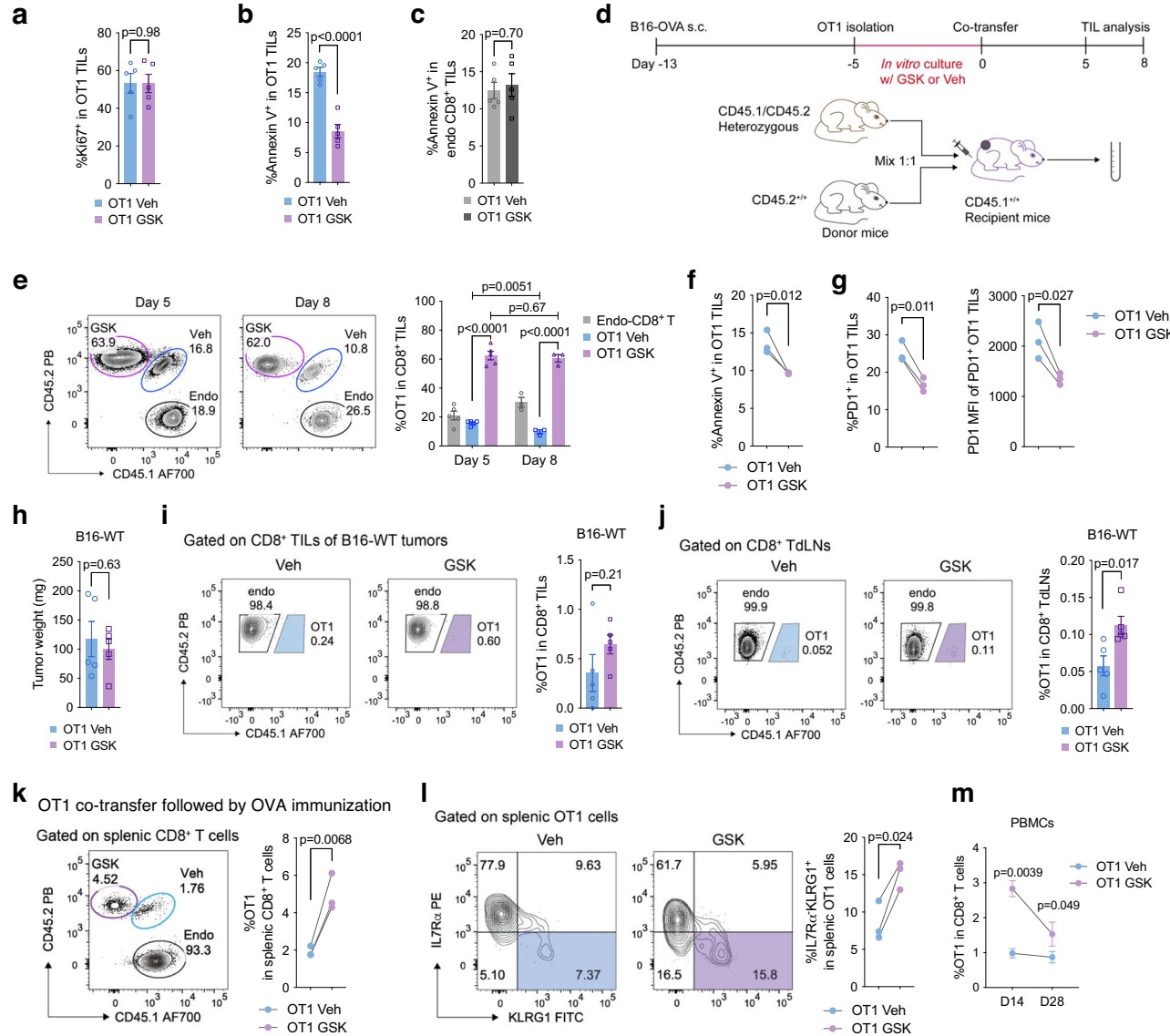

**Fig. 6 | Transient priming with GSK improves the in vivo persistence of OT1 cells.** Percentages of Ki-67⁺ (**a**) and Annexin V⁺ (**b**) cells in transferred OT1 TILs or endogenous CD8⁺ TILs (**c**) isolated from B16-OVA tumors and analyzed by flow cytometry on day 5 post transfer (*n* = 5 mice per group). **d** Experimental design for co-transfer of Veh- and GSK-primed OT1 cells to B16-OVA tumor-bearing mice and TIL analysis by flow cytometry. **e** Frequencies of Veh-primed CD45.1/CD45.2 and GSK-primed CD45.2 OT1 cells in B16-OVA tumors analyzed by flow cytometry on day 5 (*n* = 5 mice per group) and day 8 (*n* = 3 mice per group) after co-transfer. Flow cytometry analysis of percentages of Annexin V⁺ cells (**f**) and PD-1 expression (**g**) in OT1 TILs in the co-transfer assay (*n* = 3 mice per group). **h** Tumor weights of B16-WT tumor-bearing mice transferred with GSK- or Veh-primed OT1 cells (*n* = 5 mice per group). Frequencies of GSK- or Veh-primed OT1 cells in B16-WT tumors (**i**) and tumor-draining lymph nodes (TdLNs, **j**) analyzed by flow cytometry on day 5 post transfer (*n* = 5 mice per group). Frequencies (**k**) and phenotypes (**l**) of Veh-primed CD45.1/CD45.2 and GSK-primed CD45.2 OT1 cells in the spleens of mice immunized with OVA₂₅₇₋₂₆₄/poly(I:C) on day 1 and analyzed by flow cytometry on day 5 after co-transfer (*n* = 3 mice per group). **m** Frequencies of GSK- or Veh-primed OT1 cells in peripheral blood of OVA₂₅₇₋₂₆₄/poly(I:C)-immunized mice, analyzed by flow cytometry on day 14 and day 28 after co-transfer (*n* = 3 mice per group). Data in this figure are presented as mean ± SEM and represent three (**a–c**) or two (**e–g**, **k–m**) independent experiments. Statistical significance was determined by two-sided unpaired *t*-test (**a–c**, **h–j**) or paired *t*-test (**e–g**, **k–m**). Source data are provided as a Source Data file.

IL-2 has been suggested to promote T-cell exhaustion[32], which was consistently observed in our study. We showed that GSK suppressed IR expression in an IL-2 signaling-dependent manner, which in part was attributable to the inducible effect of GSK and IL-2 on EOMES expression[33]. The observation that EOMES inhibited PD-1 expression is seemingly inconsistent with previous reports, in which high levels of EOMES are associated with T-cell exhaustion[40,41]. However, EOMES has also been suggested to suppress IR expression in CD8⁺ T cells elsewhere[42,43]. It is worth noting that those in vitro activated CD8⁺ T (Teff) cells expressed low levels of PD-1, in contrast to high levels of PD-1 expression in terminal Tex cells. EOMES is possibly involved in different transcriptional networks in Teff cells compared with Tex cells[44], thus regulating PD-1 expression in a context-dependent way.

In summary, our study reveals a role of LSD1 in regulating the phenotype and function of adoptively transferred T cells and raises an approach to targeting LSD1 to enhance the antitumor efficacy of ACT therapy. Our findings that a transient and timely inhibition of LSD1 is sufficient for therapeutic enhancement are instructive for the potential inclusion of LSD1 pharmacological inhibitors in ACT therapy.

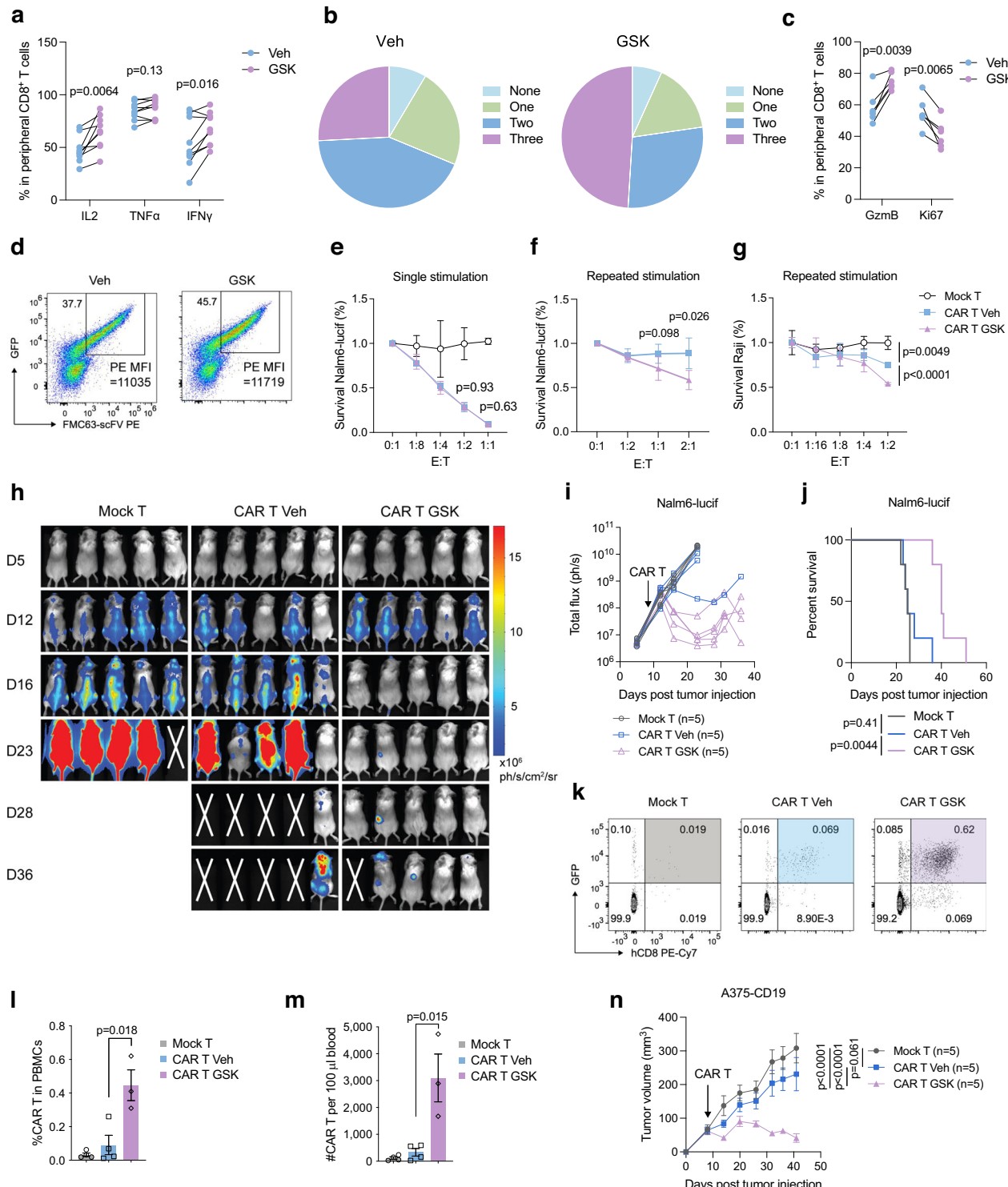

**Fig. 7 | Pharmacological LSD1 inhibition enhances the antitumor potency of human CAR T cells.** Flow cytometry analysis of cytokine production (**a**) and multiple-cytokine producers (**b**) in human peripheral CD8[+] T cells activated and expanded in vitro for 10 days with or without GSK treatment (*n* = 9 donors). **c** Flow cytometry analysis of GzmB and Ki-67 expression in GSK- or Veh-treated human CD8[+] T cells (*n* = 6 donors). **d** Flow cytometry analysis of FMC63-scFv and EGFP expression in human CD8[+] T cells transduced with a lentiviral plasmid encoding CD19-CAR and EGFP and treated with or without GSK. Cytotoxicity of GSK- or Veh-treated CD19-CAR T cells or unmodified mock T cells in tumor killing assays using CD19[+] Nalm6-lucif-EGFP (**e, f**) or Raji (**g**) cells as targets and single-stimulated (**e**) or repeatedly stimulated (**f, g**) CD19-CAR T cells as effectors (*n* = 3 for **e** and **g**, and *n* = 4 for **f**). Representative bioluminescence images (**h**), quantitative bioluminescence imaging data (**i**), and survival curves (**j**) of NCG mice intravenously injected

with 10[6] Nalm6-lucif-EGFP cells and infused with 2 × 10[5] CD19-CAR or mock T cells 7 days later (*n* = 5 mice per group). Representative flow plots (**k**), frequencies (**l**), and cell numbers (**m**) of human mock CD8[+] T cells and GSK- or Veh-primed CD19-CAR CD8[+] T cells in peripheral blood of NCG recipient mice on day 12 post transfer (*n* = 4 mice for mock T and CAR T Veh, and *n* = 3 mice for CAR T GSK). **n** Tumor growth curves of NCG mice subcutaneously injected with 10[6] A375-CD19 cells and infused with 2 × 10[5] CD19-CAR T or mock T cells 8 days later (*n* = 5 mice per group). Data are pooled from three independent experiments (**a–c**) or represent two independent experiments (**d–m**) and are presented as mean ± SD (**e–g**) or mean ± SEM (**l–n**). Statistical significance was determined by paired *t*-test (**a, c**), two-sided unpaired *t*-test (**e–g, l, m**), two-way ANOVA (**n**), or log-rank test (**j**). Source data are provided as a Source Data file.

## Methods

### Ethics statement

All animal experiments were performed in accordance with the animal care guidelines and with the prior approval by the Zhejiang University School of Medicine Institutional Animal Care and Use Committee. Peripheral blood samples were collected from volunteers with the written informed consent, with approval by the Clinical Research Ethics Committee of the First Affiliated Hospital, College of Medicine, Zhejiang University (2023-0349).

### Cell culture

Mouse CD8[+] T cells were cultured in RPMI 1640 medium supplemented with 10% FBS, 50 μM β-mercaptoethanol (Macklin, M861503), and 20 ng/mL recombinant human IL-2 (Novoprotein, C013). Human PBMCs were cultured in X-Vivo$^{TM}$ 15 serum-free medium (Lonza, 02-053Q) supplemented with 50 ng/mL IL-2. HEK293T, B16-OVA, Yumm1.7-OVA, and A375 cells were cultured in DMEM medium supplemented with 10% FBS. EL4-OVA, EL4, Nalm6-lucif-EGFP, and Raji cells were cultured in RPMI 1640 medium supplemented with 10% FBS. All cells were cultured in a 5% $CO_2$ incubator at 37 °C and detected to be mycoplasma-free by a PCR-based method[45].

### Mice

Six-to-eight-week-old female C57BL/6 mice were purchased from the Shanghai SLAC Laboratory Animal Co., Ltd. Six-to-seven-week-old female NCG (NOD/ShiLtJGpt-Prkdc$^{em26Cd52}$Il2rg$^{em26Cd22}$/Gpt) mice (T001475) were purchased from GemPharmatech Co., Ltd. Lsd1$^{flox/flox}$ mice were generously provided by Dr. Weiguo Zou at Center for Excellence in Molecular Cell Science, CAS. Stat5$^{flox/flox}$ mice were generously provided by Dr. Xin-Yuan Fu at West China Hospital. OT1 (C57BL/6-Tg(TcraTcrb)1100Mjb/J, 003831), CD45.1 (B6.SJL-Ptprc$^a$Pepc$^b$/BoyJ, 002014), Cd4-Cre (Tg(Cd4-cre)1Cwi/BfluJ, 017336), Rosa26-Cas9 (B6J.129(Cg)-Gt(ROSA)26Sor$^{tm1.1(CAG-cas9*,-EGFP)Fezh}$/J, 026179), and Rosa26-CreERT2 (B6.129-Gt(ROSA)26Sor$^{tm1(cre/ERT2)Tyj}$/J, 008463) mice were originally purchased from The Jackson Laboratory. Prior to all experiments, purchased mice were housed for one week to acclimate to the conditions at the Zhejiang University Laboratory Animal Center. All experimental mice were housed in specific pathogen-free conditions with a 12-h light/12-h dark cycle and controlled temperature (-22 °C). Animal experiments were performed in accordance with the animal care guidelines and with the prior approval by the Zhejiang University School of Medicine Institutional Animal Care and Use Committee.

### Mouse CD8[+] T-cell isolation, activation, and lentiviral transduction

Mouse CD8[+] T cells were isolated from the spleens of mice with an EasySep™ Mouse CD8[+] T Cell Isolation Kit (STEMCELL Technologies, 18953) according to the manufacturer's manual. CD8[+] T cells were stimulated with 2 μg/ml plate-bound anti-CD3 (BioLegend, 100340) and 2 μg/ml soluble anti-CD28 (BioLegend, 102116) with the addition of 20 ng/mL IL-2 at a density of 1 million/ml for 48 h. Activated CD8[+] T cells were then expanded with IL-2 in fresh medium for the indicated time before analysis. 10 μg/ml anti-IL-2 (BioLegend, 503706), 0.04 μM tofacitinib (MedChem Express, HY-40354), 100 nM 4-hydroxytamoxifen (MedChem Express, HY-16950), 0.05–1 μM GSK2879552 (MedChem Express, HY-18632), 0.05–1 μM ORY1001 (Cayman, 19136), 10–50 nM decitabine (MedChem Express, HY-A0004), 0.1–1 μM GSK126 (MedChem Express, HY-13470), 2–20 nM chaetocin (MedChem Express, HY-N2019), or vehicle control was added as indicated.

For lentiviral production, HEK293T cells were co-transfected with the lentiviral plasmid (lentiCRISPRv2 puro or lenti-FMC63-BBz-GFP), packaging plasmid psPAX2 (Addgene, 12260), and envelope plasmid pMD2.G (Addgene, 12259), followed by viral supernatant collection at 72 h. The lentiviral particles were used immediately or stored at -80 °C

for later use. For lentiviral transduction, isolated T cells were activated with anti-CD3/anti-CD28 for 24 h. Lentiviral supernatant was directly added into a plate well containing pre-activated T cells at a 1:1 (v/v) ratio with the addition of 6 μg/ml polybrene (Solarbio, H8761), followed by centrifugation at 2000 × g for 90 min at 30 °C. Afterward, the supernatant was removed and T cells were cultured with IL-2 in fresh medium at a density of 1 million/ml. For transduction of Rosa26-Cas9 CD8[+] T cells with lentiCRISPRv2 puro, the following guide RNA oligos were used: TTCTGGCCGACCCTAACCAC (Eomes) and GCACTACCAGA GCTAACTCA (Scramble).

### Protein extraction and western blot analysis

Cells were collected and lysed in RIPA lysis buffer (Beyotime, P0013C) supplemented with the complete protease inhibitor cocktail (Yeasen, 20124ES03) on ice for 10 min. After centrifugation at 20,000 × g for 10 min at 4 °C, supernatant was collected and protein concentrations were quantified with a Bradford Protein Assay Kit (Coolaber, SK1060). After adding loading buffer (Beyotime, P0015L) and boiling at 98 °C for 5 min, protein extracts were resolved on an SDS-PAGE gel and then transferred to a nitrocellulose membrane (Pall, 66485). The membrane was blocked with 5% skim milk at room temperature for 1 h, followed by incubation with primary antibodies at 4°C overnight. The following antibodies were used at 1:1000 dilution: Phospho-Stat5 (Tyr694) (Cell Signaling Tech, 9359), Phospho-PI3 Kinase p85 (Tyr458)/p55 (Tyr199) (Cell Signaling Tech, 4228), Phospho-p44/42 MAPK (Erk1/2) (Thr202/Tyr204) (Cell Signaling Tech, 9101), p44/42 MAPK (Erk1/2) (Cell Signaling Tech, 4695), and LSD1 (Santa Cruz Biotech, sc-53875). After washing the membranes with PBST (PBS, 0.1% Tween-20) three times, the membrane was incubated with HRP-conjugated goat anti-rabbit IgG (Biosharp, BL003A) or HRP-conjugated goat anti-mouse IgG (Biosharp, BL001A) diluted in 5% skimmed milk at room temperature for 1 h. After three washes with PBST, ECL was applied to the membranes and imaged using Tannon-5200.

### The cellular thermal shift assay

In vitro activated and expanded CD8[+] T cells were treated with GSK2879552 or ORY1001 at the indicated concentrations and for the indicated times. Cells were collected and washed once with PBS at room temperature. For the apparent melting curve experiments, cells were resuspended in PBS supplemented with the protease inhibitor cocktail and aliquoted into eight 0.2-ml strip PCR tubes, each containing 2 million cells in a volume of 100 μl. Eight PCR tubes were heated in a PCR cycler for 3 min at eight temperature endpoints (42, 45, 48, 51, 54, 57, 60, and 63 °C) respectively. For the dose-response experiments and time-duration experiments at a fixed temperature, 2 million cells from each treatment condition were resuspended in 100 μl PBS and transferred into a 0.2-ml strip PCR tube, followed by heating in a PCR cycler for 3 min at 53 °C.

Immediately after heating, cell suspensions were incubated at 25 °C for 3 min, followed by two cycles of snap-freezing in liquid nitrogen and thawing in a water bath at 25 °C. The tubes were vortexed briefly after each thawing. The resulting cell lysates were centrifuged at 20,000 × g for 10 min at 4 °C, and 80 μl of each supernatant was collected for immunoblot analysis of LSD1. LSD1 protein levels were quantified to establish the apparent melting curves and dose-response curves.

### CFSE proliferation assay

Isolated CD8[+] T cells were suspended in PBS at a concentration of $10 × 10^6$/ml and labeled with a Celltrace™ CFSE Cell Proliferation Kit (ThermoFisher Scientific, C34570) at 37 °C for 20 min protected from light. After washing with RPMI 1640 culture medium, cells were resuspended in fresh culture medium. CFSE-labeled T cells were stimulated with anti-CD3/anti-CD28 and IL-2 for 72 h in the presence of 0.5 μM GSK2879552, ORY1001, or vehicle. After the addition of

7-AAD, cells were analyzed on a BD LSRFortessa for CFSE signal detection.

## Calcium influx assay

In vitro cultured CD8$^+$ T cells were incubated with 5 μM fluorescent dye Fluo-4 AM (Thermo Fisher Scientific, F14201) for 30 min in PBS at 37 °C. After washing with PBS, cells were labeled with PE-conjugated anti-CD8a (BioLegend, 300908) and Zombie Aqua™ dye (BioLegend, 423101) for 15 min and resuspended in PBS after washing. PMA and CaCl2 were sequentially added to cells while running on an ACEA NovoCyte™. Flow data were analyzed with the FlowJo (10.4).

## Mouse subcutaneous tumor models and adoptive T-cell transfer

Mice were anesthetized with isoflurane, shaved at the right hind flank, and subcutaneously injected with 0.5 million B16-OVA or Yumm1.7-OVA cells per mouse. 8–10 days after tumor injection, mice were randomized to different treatment groups and transferred with in vitro activated and expanded OT1 cells (1 or $5 \times 10^6$ per mouse as indicated) intravenously via tail vein. The following epigenetic inhibitors were used to prime OT1 cells: 0.5 μM GSK2879552, 0.5 μM ORY1001, 10 nM decitabine, 1 μM GSK126, and 5 nM chaetocin. For PD-1 blockade therapy, 100 μg anti-PD-1 (BioXCell, BE0146) or IgG2a isotype control (BioXCell, BE0089) were injected intraperitoneally every 2 days as indicated. Tumor long (D) and short (d) diameters were measured with a digital caliper every 2–3 days and were used to calculate tumor volumes by the formula: $1/2 \times D \times d^2$. Mice were sacrificed when tumors reached 2000 mm³ or upon ulceration or bleeding. In some cases, this limit has been exceeded the last day of measurement and the mice were immediately euthanized.

For co-transfer assays, CD45.1 mice were used as recipients for tumor implantation. OT1 cells were isolated from CD45.1/CD45.2 heterozygous or CD45.2 mice, activated with anti-CD3/anti-CD28 for 2 days and expanded with IL-2 for 3 days, throughout which vehicle control or 0.5 μM GSK2879552 was added respectively. After in vitro activation and expansion, two groups of OT1 cells were mixed in equal numbers and transferred to tumor-bearing CD45.1 mice (a total of 2 million cells per mouse).

## Tumor-infiltrating leukocyte analysis and flow cytometry

Tumors were collected on day 5 or day 8 after adoptive T-cell transfer and cut into 2 mm-sized pieces in RPMI1640 medium with the addition of type I collagenase (Worthington Biochemical Corporation, LS004194) and DNase I (Sigma-Aldrich, 10104159001). Tumor tissues were then digested at 37 °C for 20–30 min and passed through a 70 μm cell strainer to obtain a single-cell suspension. To enrich leukocytes, samples were spun through a Percoll (GE Healthcare Life Sciences, 17-0891-01) gradient for 20 min at 2000 rpm without brake. Tumor-infiltrating leukocytes were collected from the interface of the 40% and 70% Percoll gradient, stained, and analyzed for fluorescent markers.

For surface staining, cells were blocked with anti-CD16/32 for 5 min and then stained for 30 min at 4 °C with the flowing antibodies at 1:200 dilution: CD8a BV605 (BioLegend, 100744), TCRβ BV510 (BioLegend, 109233), CD45.1 AF700 (Biolegend, 110724), CD45.2 PB (BioLegend, 109820), PD-1 PE/Cyanine7 (BioLegend, 135215), TIM-3 BV711 (BioLegend, 119727), TIM-3 APC (BioLegend, 119705), CD39 PE (BioLegend, 143803), CD44 APC (BioLegend, 103012), CD62L FITC (BioLegend, 104405), CD62L Pacific blue (BioLegend, 161207), IL7Rα PE (BioLegend, 135009), KLRG1 FITC (BioLegend, 138409), CD25 APC (BioLegend, 102011), and SLAMF6 APC (BioLegend, 134609). For intracellular proteins, after surface staining and washing, cells were fixed and permeabilized with a Foxp3 Fixation/Permeabilization kit (Thermo Fisher Scientific, 00-5523-00), and stained with the following antibodies at 1:200 dilution: Ki-67 PerCP/Cyanine5.5 (BD Pharmingen, 561284), Granzyme-B AF647 (BioLegend, 515405), EOMES PE (ThermoFisher Scientific, 12-4875-80), and LSD1 (Abcam, ab17721). For

cytokine staining, cells were stimulated in RPMI 1640 culture medium with PMA (Cayman, 10008014), ionomycin (Cayman, 10004974), and Golgiplug (BD Pharmingen, 555029) for 4 h, and subjected to surface staining before being fixed and permeabilized with a Cytofix/Cytoperm™ Kit (BD Pharmingen, 554714). The following antibodies were used at 1:200 dilution: IL-2 FITC (BioLegend, 503805), TNFα FITC (BioLegend, 506303), and IFNγ PE (BioLegend, 505807). 7-AAD (BioLegend, 420404) or Zombie NIR™ dye (BioLegend, 423106) was used to exclude dead cells. Data were acquired on a BD LSRFortessa and analyzed using FlowJo (10.4).

## ELISA

The ELISA assay was performed with an IL-6 Mouse ELISA Kit (Invitrogen, 88-7064-22) and IL-1 beta Mouse ELISA Kit (Invitrogen, 88-7013-22) according to the manufacturer's instructions. Serum was collected from B16-OVA tumor-bearing mice two weeks after the transfer of 5 million GSK- or Veh-primed OT1 cells, or PBS.

## Human PBMC isolation, culture, and CD19-CAR T-cell generation

Peripheral blood samples were obtained from healthy volunteers approved by the Clinical Research Ethics Committee of the First Affiliated Hospital, College of Medicine, Zhejiang University (2023-0349). Informed consent from all participants was obtained. Peripheral blood mononuclear cells (PBMCs) were isolated from the blood by density gradient centrifugation with Ficoll. T cells from PBMCs were activated with human T-cell activation beads (GenScript, L00899) in X-Vivo 15 medium at a density of 1 million/ml for 2 days and then expanded with 50 ng/ml IL-2 for several days, throughout which 2 μM GSK2879552 or vehicle was added.

The CD19-specific CAR construct containing FMC63 scFv, 4-1BB cytoplasmic domain, and CD3z cytoplasmic domain was obtained from Addgene (135992) and cloned into an EGFP-expressing lentiviral vector. Lentiviral particles were produced as described above and used to transduce pre-activated PBMCs by spin-infection. The expression of CD19-CAR was verified by flow cytometry after the sequential staining with a biotin-SP-AffiniPure F(ab)'2 fragment-specific goat anti-mouse IgG antibody (Jackson ImmunoResearch, 115-066-072) and streptavidin-phycoerythrin (BioLegend, 405203). Transduced cells were cultured and expanded in X-Vivo 15 medium supplemented with 50 ng/mL IL-2 for 6–8 days before in vitro assays or transfer into tumor-bearing NCG mice.

## Flow cytometry analysis of human T cells

For surface staining, in vitro cultured human T cells or NCG mice-retrieved PBMCs were stained for 30 min at 4 °C with the following antibodies at 1:200 dilution: CD8a PE/Cyanine7 (BioLegend, 344711), PD-1 PB (BioLegend, 329915), TIM-3 FITC (BioLegend, 345021), and CD39 PE (BioLegend, 328207). Intracellular staining was performed as described above and the flowing antibodies were used at 1:200 dilution: TNFα APC (BioLegend, 502913), IFNγ FITC (BioLegend, 502505), IL-2 PE (BioLegend, 500306), Ki-67 PerCP/Cyanine5.5 (BD Pharmingen, 561284), and Granzyme-B AF647 (BioLegend, 515405). Samples were run on a BD LSRFortessa and data were analyzed using FlowJo (10.4).

## Repeated stimulation assay

OT1 cells were activated with anti-CD3/anti-CD28 and IL-2 for 48 h, and then repeatedly stimulated with 2 μg/ml plate-bound anti-CD3 and 20 ng/ml IL-2 every other day for a total of 4 rounds. As a control, after 48 h activation, a fraction of cells were expanded with IL-2 only. The phenotype and effector function of those cells were analyzed by flow cytometry and T-cell killing assays.

Human PBMCs after lentiviral transduction with CD19-CAR were plated into a well pre-coated with 10 μg/ml anti-CD3 (BioLegend 317326) for 48 h with the addition of 50 ng/ml IL-2. The stimulation was repeated 2–3 times before cells were used in T-cell killing assays.

### OT1 and EL4-OVA co-culture assay

EL4-OVA and EL4 tumor cells were equally mixed and seeded in RPMI 1640 culture medium one day prior to co-culture. On the next day, pre-activated OT1 cells were added to the tumor cells at a 1:1 ratio supplemented with 20 ng/ml IL-2. Subsequent co-cultures were repeated every 48 h for a total of 2 rounds. Afterward, the phenotype and effector function of OT1 cells were analyzed by flow cytometry.

### In vitro T-cell killing assay

Antigen-specific T cells after the indicated treatment were co-cultured with pre-plated tumor cells in a 96-well flat bottom plate at serially diluted $E/T$ ratios in triplicates for an appropriate time (~18 h). B16-OVA and B16-GFP cells were 1:1 mixed and used as target tumor cells for OT1 cell killing assay. The antigen-dependent killing effect was evaluated by ratios of %B16-OVA over %B16-GFP, analyzed by flow cytometry gated on the CD45$^-$ population, and normalized to the group of $E/T = 0$. CD19$^+$ Nalm6-lucif-EGFP and Raji cells were used for the CD19-CAR T-cell killing assay. The number of live Nalm6-Lucif-EGFP or Raji cells was counted by flow cytometry and normalized to the group of $E/T = 0$ to obtain the survival percentages.

### In vivo xenograft tumor models and adoptive CAR T-cell transfer

For the leukemia tumor model, after acclimation, female NCG mice were intravenously injected with 1 million Nalm6-lucif-EGFP cells via the tail vein. After 7 days, mice were randomly divided into three groups, followed by intravenous injection of $2 \times 10^5$ Veh- or GSK-primed CD19-CAR T cells, or non-transduced mock T cells. Leukemia burdens were evaluated weekly using a bioluminescence imaging system (Optima, Biospace Lab) after the intravenous administration of D-Luciferin (Yeasen, 40902ES03). Mouse peripheral blood was collected to analyze the abundance of CD19-CAR T cells by flow cytometry on day 12 after adoptive transfer.

For the melanoma tumor model, female NCG mice were anesthetized with isoflurane, shaved at the right hind flank, and subcutaneously injected with 1 million A375-CD19 cells per mouse. Eight days after tumor injection, mice were randomized into three groups and infused with $2 \times 10^5$ Veh- or GSK-primed CD19-CAR T cells, or non-transduced mock T cells. Tumor volumes were measured and calculated as described above. Mice were sacrificed when tumors reached 1000 mm$^3$ or upon ulceration or bleeding.

### RNA-seq and analysis

In vitro cultured CD8$^+$ T cells under the indicated treatment were directly lysed in 0.5 ml TRIzol reagent (TaKaRa, 9109) and incubated at RT for 5 min. 1/5 volume of chloroform was added and mixed gently by inversing tubes several times. Cell lysates were incubated for 3 min at room temperature before centrifugation at $12,000 \times g$ for 15 mins at 4 °C. RNA in the upper aqueous phase was transferred into a new tube and 1:1 mixed with freshly prepared 70% ethanol. The mixture was then loaded into a column from an RNA Clean & Concentrator™-5 Kit (ZYMO Research, R1013), followed by RNA purification according to the manufacturer's instructions. Purified RNA was quantified using a NanoDrop 2000 Spectrophotometer (Thermo Scientific) and assessed for RNA integrity using an Agilent 2100 Bioanalyzer. Total RNA with RIN > 9.0 was used for the poly(A)$^+$ RNA enrichment, which was then subjected to library construction using a TruSeq Stranded mRNA Sample Preparation Kit (Illumina, RS-122-2101) according to the manufacturer's instructions. Prepared libraries were sequenced at paired-end 150 bp on an Illumina Novaseq 6000 platform, generating over 40 million reads per sample.

Raw reads were processed using Trimmomatic to remove adapters and low-quality reads, and trim low-quality bases. Clean reads were mapped to the mouse GRCm38 genome using hisat2 (v2.1.0). Read counts per gene were generated using HTSeq-count (v0.11.2).

FPKM were calculated using Cufflinks (v2.2.1) to estimate gene expression levels. Differential gene expression was analyzed using R package DESeq2 (v1.22.2), with the cutoff set at fold change > 1.5 and $q$-value < 0.05. The $p$ values were adjusted for multiple testing using the Benjamini-Hochberg method and an FDR adjusted $p$-value (or $q$-value) <0.05 was considered statistically significant. For gene ontology (GO) enrichment analysis, the differentially expressed genes were queried to the Gene Ontology Consortium using R package clusterProfiler (v3.0.4), with the specification of biological process.

### ChIP-seq and analysis

In vitro cultured CD8$^+$ T cells were fixed with 1% formaldehyde (Thermo Fisher Scientific, 28908) for 10 min at room temperature on an orbital shaker and then quenched with 125 mM glycine for 10 min. After washing twice with ice-cold PBS, cells were lysed in sonication buffer (140 mM NaCl, 50 mM HEPES pH7.9, 1 mM EDTA, 1% Triton X-100, 0.1% sodium deoxycholate, 0.2% SDS, and protease inhibitor) and sonicated to generate DNA fragments enriched at 200–800 bp. Subsequent procedures were done following the Epigenetics Protocols Database PROT-11 (https://www.epigenome-noe.net/researchtools/protocols.php.html). For immunoprecipitation, the following antibodies were used: anti-LSD1 (Abcam, ab17721), anti-Pol II (Santa Cruz Biotechnology, sc-56767), anti-H3K27ac (Active Motif, 39034), anti-H3K4me1 (Abcam, ab8895), anti-H3K4me2 (EMD Millipore, 07-030), anti-H3K4me3 (Active Motif, 39060), and rabbit normal IgG (Proteintech, 30000-0-AP). ChIP-seq libraries were prepared using the VAHTS Universal DNA Library Prep Kit (Vazyme, ND607-01) according to the manufacturer's instructions. Library concentrations were quantified by Qubit (Invitrogen) and mixed equally for sequencing at Novaseq 6000 (Illumina) to generate 150 bp paired-end reads or at DNBSEQ-T7 (MGI Tech) to generate 100 bp paired-end reads.

ChIP-seq reads were mapped to unique genomic regions of mm10 using Bowtie2 (v2.3.5.1). PCR duplicates were removed manually using Samtools (v1.9). The bedgraph files were generated using the Bedtools (v2.27.1) genome coverage function. Libraries were normalized by randomly picking the same numbers of reads in order to compare changes in ChIP-seq signals. Normalized reads were used to generate bedgraph files for visualization in IGV. Peak calling was performed using MACS2 (v2.1.2) with default parameters and $p$-value cutoff at 0.001. Differential peaks were identified using the edgeR R package with the threshold set at $p < 0.05$ and changes over 20% up/down.

To derive heatmaps of ChIP-seq signals, a blank matrix was first generated by binning an appropriate size of regions centering on anchors equally, where each row represented one anchor and each column represented one bin. Then all datasets from the same antibody were normalized by random picking to get equal numbers of read pairs for all following analysis. To obtain signals of the whole matrix, we applied the bedtools intersect function to map the normalized read pairs derived in the previous step to each genomic bin to obtain read counts in all bins. To get sequencing depth normalized counts for the matrix, raw counts derived in the previous step were divided by library sizes in millions to get reads per million per covered bin (RPMPB) as final counts in the matrix. All matrices were visualized with Java Tree-View to derive heatmaps. Column means of the matrix were calculated to generate average profiles of the ChIP-seq signals neighboring specific anchors. To derive K-means clustering of ChIP-seq heatmaps, center ± 3 bins signals of appropriate heatmaps were used as input. For K-means clustering of promoter heatmaps, TSSs ± 3 bins signals of H3K27ac or H3K4me2 heatmaps were taken as Cluster 3 input. All heatmaps of TSSs ± 1 kb were plotted in the same order of K-means clustering.

Enhancers were defined by merging H3K4me1 and H3K4me2 peaks in both conditions but excluding H3K4me3 peaks and TSS ± 2.5 kb. To search for differential enhancers, H3K27ac ChIP-seq signals in both conditions were used as input for the edgeR R package with the

threshold of *p* value < 0.05 and more than 3-fold changes in either condition. Center ± 3 bins signals of H3K4me1 heatmaps were used as input for Cluster 3 to derive K-means clustering of enhancer heatmaps. All enhancer heatmaps on center ± 1 kb were plotted in the same order of K-means clustering.

## Statistics

Statistical analyses were performed using GraphPad Prism (version 9.2) and statistical significance was determined by *p* < 0.05. A two-sided unpaired or paired Student's t-test was used for comparisons between the two groups. A two-way ANOVA test was used for multiple comparisons of tumor growth. A Log-rank (Mantel-Cox) test was used for comparing survival curves. Statistical data from experimental, biological, or technical replicates were presented as mean ± SEM or SD. The number of replicates and the statistical test used were indicated in the corresponding figure legends.

## Reporting summary

Further information on research design is available in the Nature Portfolio Reporting Summary linked to this article.

## Data availability

The RNA-seq and ChIP-seq data generated in this study have been deposited in the Gene Expression Omnibus (GEO) database under accession codes: GSE248892 and GSE248891. The remaining data are available within the Article, Supplementary Information, or Source Data file. Source data are provided with this paper.

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

## Acknowledgements

We thank Dr. Dante Neculai for critical reading of the manuscript and valuable suggestions. We thank Drs. Weiguo Zou, Xinjun Zhang, Xin-Yuan Fu, Lie Wang, and Linrong Lu for sharing mouse strains; Drs. Zhuang Liu, He Huang, Bin Zhao, and Zhijian Cai for sharing cell lines; Dr. Dongrui Wang for sharing cDNA plasmids and cell lines; Yihan Yao, Ying Shen, Jiali Tao, Chenyang Lu, and Nan Jiang for technical help. We thank Yanwei Li from the Core Facilities of Zhejiang University School of Medicine for assistance with flow cytometric analysis and the Laboratory Animal Center of Zhejiang University for the animal service. This work was supported by funds from the National Key R&D Program of China (2022YFC3401600 and 2022YFA1104900 to W.S., 2021YFC2700300 and 2021YFC2700303 to X.L.), the National Natural Science Foundation of China (32170907 and 32370964 to W.S., 81972179 to J.Z., 32170546 to X.L.), the Key R&D Program of Zhejiang Respiratory Disease (2023C03069 to J.Z.), Zhejiang University School of Medicine, and Liangzhu Laboratory.

## Author contributions

W.S., F.Q., and P.J. designed the experiments. F.Q. and P.J. performed most experiments and analyzed the data with assistance from G.Z., J.A., and K.R. under the supervision of W.S. and J.Z. X.L. analyzed the ChIP-seq data. W.S., P.J., and F.Q. wrote the manuscript. All authors discussed the results and commented on the manuscript.

## Competing interests

The authors declare no competing interests.
