## [Peer Review File · Nature Communications]

Priming with LSD1 inhibitors promotes the persistence and antitumor effect of adoptively transferred T cellsREVIEWER COMMENTS

Reviewer #1 (Remarks to the Author): with expertise in cancer immunology, T-cell therapies

The authors show that targeting histone demethylase LSD1 with chemical inhibitors reshapes the epigenome of activated T cells and potentiates their antitumor activity. The authors claim that LSD1 inhibition leads to a higher percentage of cells with a memory phenotype, higher ability to produce cytokine, lower frequency of exhaustion markers, and higher persistence. The authors also show an increase in the antitumor activity of OT-1 cells against B16-OVA tumors alone and in combination with anti-PD1. The manuscript is interesting and well-written. However, some major points need to be addressed before publication.

1. Antitumor activity of LSD1 inhibition is shown only in a murine model of OT-1 T cells and B16-OVA or Yumm1.7-OVA. It is relevant to show the activity of LSD1 inhibition in additional human adoptive T cell therapy models. A relevant model could be human CD19CAR-T cells, among others.

2. The field is discovering several similar approaches to increase the antitumor activity of adoptively transferred T cells based on epigenetic modifications. These approaches lead to similarly reported increases in T cell antitumor activity; however, it would be very relevant for the field to start benchmarking these approaches against each other. I would suggest comparing the epigenetic modifications and the antitumor activity obtained with LSD1 inhibitors and other epigenetic modulators such as DNMT3a inhibitors, decitabine, and EZH2 and SUV39H1 inhibitors.

3. A potential concern of this treatment is the antigen-independent release of cytokine that could lead to severe toxicities of this treatment. In Figure 1 (murine T cells) and Figure 7 (human T cells), T cells are activated for 48h and then expanded with and without drug and IL2. After expansion, these cells show higher levels of cytokines and granzyme without antigen stimulation that could lead to toxicities once administered systemically. Please expand on this issue and justify if this is not a concern. Toxicology data in vivo would be

required to confirm the potential lack of toxicity.

Minor comments:

Figure 2a. The lower part (Defined enhancers) is missing all the titles. It would be easier to read if added, like in the top part of the figure.

Line 334. I would suggest using “not statistically significant” instead of “statistically insignificant”.

Reviewer #2 (Remarks to the Author): with expertise in cancer, epigenetics, LSD1

This is a strong paper showing that transient LSD1 inhibition can augment adoptive T cell therapy. Most experiments are conducted in mouse T cells models and include melanoma tumor bearing models, and a final experiment uses human T cells in a Nalm6 model. Some data is overstated and requires additional validation as listed below:

1. Rationale for the dose of GSK2879552 and ORY1001 used in Figure 1 and supplementary Figure 1 should be provided. In particular, authors should show whether LSD1 is actually inhibited in T cells at these doses. This could be accomplished by CETSA.
2. Supplementary Figure 2a should be carried out with ORY1001 to confirm the survival advantage of activated CD8+ cells is seen with another inhibitor.
3. Some explanation for the finding in Figure 2a should be offered regarding why H3K4me1 and H3K4me2 are unchanged. What are the putative genes that LSD1 is indirectly regulating?
4. Figure 3 and supplementary figure 4 should show EOMES depletion by western or flow.
5. A major finding of this study is in Figure 5 - showing that GSK priming duration is critical for efficacy. Can the authors correlate this with degree of kinetics and extent of LSD1

inhibition.

6. Similar question for figure 6 - what is a molecular readout of adequate LSD1 priming?

7. In Figure 7, why was the NALM6 model used for the human CAR T cells and not a melanoma model as done in the prior mouse expts?

Reviewer #3 (Remarks to the Author): with expertise in cancer immunology

The manuscript by Qiu et al combines chemical inhibitor and genetic knockout data to reveal the potential role and relevant mechanism of LSD1 in suppressing adoptive CD8+T cell therapy. In particular, the authors report that the inhibition of LSD1 reshapes the epigenome and improves the memory phenotype of both murine and human CD8+T cell, thus enhancing the antitumor effect in solid tumor models after adoptive transfer alone or in combination with PD1 blockade.

Collectively, this is an interesting and comprehensive study that increases our understanding of T cell exhaustion regulation, especially PD1 regulation, and provide evidence for combined trial of LSD1 inhibition and PD1 blockade in the context of adoptive CD8+T cell therapy.

However, not all proposed mechanisms are sufficiently supported by the data.

Major comments:

1. For Fig. 1j, besides the cell death maker 7AAD, the authors should also check the expression of the proliferation maker Ki67.
2. The data in Fig. 3a-3b does not support the conclusion in line 183 that, GSK treatment inhibits TCR activation-induced PD1 expression. Please revise and do not over-interpret data.
3. For Fig. 5j-5k, the authors should also add a "GSK+IgG" group to demonstrate the tumor suppression effect of "GSK+anti-PD1" is superior to GSK treatment alone.
4. Since there are a lot of grammar errors and misuse of tense in the manuscript, the authors had better ask for help from a professional or a native English speaker to correct the errors.

Minor comments:

1. For representative data in Fig. 1f, the authors should show the frequency and the way they set the gates for IL-2+, TNFa+ and IFNr+ cells.
2. For representative data in Fig. 4j-4l, the authors should mark the treatment type accordingly.

We would like to thank the reviewers for their constructive comments, which have helped us improve our manuscript significantly. Our key new data are summarized below to address the editor's particular comments, followed by a detailed response to the reviewers' comments.

- 1) To justify the rationale for the dose of LSD1 inhibitors used in this study, we have analyzed the phenotypes of CD8⁺ T cells in response to GSK2879552 and ORY1001 treatment at a range of concentrations (50–1000 nM), which showed that the selected dose (0.5 μM) was sufficient to produce a maximum effect without reducing cell survival or proliferation. Further, we have performed the CETSA assay, as suggested by Reviewer #2, which confirmed the complete inhibition of LSD1 in T cells at the selected dose of inhibitors.
- 2) To validate our main findings in additional models, we have used human CD19-CAR T cells to treat mice carrying Nalm6 leukemia or A375 melanoma (transduced with CD19), and found that GSK2879552 priming significantly enhanced the antitumor efficacy of adoptively transferred CAR T cells in both models, in line with the findings from the OT-1 transfer model.
- 3) To provide benchmarking for epigenetic modulators in adoptive T cell therapy, we have compared LSD1 inhibitors (GSK2879552 and ORY1001) with several reported epigenetic modulators, including decitabine (a DNMT inhibitor), GSK126 (an EZH2 inhibitor), and chaetocin (a SUV39H1 inhibitor), in terms of their effects on T cell activation and antitumor activity. As a result, GSK2879552 performed similarly to chaetocin but better than decitabine and GSK126 at enhancing the antitumor efficacy of OT1 cells in the B16-OVA tumor model, highlighting the effectiveness of LSD1 inhibition in improving adoptive T cell therapy.

Please see below, in **blue**, for a point-by-point response to the reviewers' comments.

Responses to Reviewers' Comments

Reviewer #1 (Remarks to the Author): with expertise in cancer immunology, T-cell therapies

The authors show that targeting histone demethylase LSD1 with chemical inhibitors reshapes the epigenome of activated T cells and potentiates their antitumor activity. The authors claim that LSD1 inhibition leads to a higher percentage of cells with a memory phenotype, higher ability to produce cytokine, lower frequency of exhaustion markers, and higher persistence. The authors also show an increase in the antitumor activity of OT-1 cells against B16-OVA tumors alone and in combination with anti-PD1. The manuscript is interesting and well-written. However, some major points need to be addressed before publication.

1. Antitumor activity of LSD1 inhibition is shown only in a murine model of OT-1 T cells and B16-OVA or Yumm1.7-OVA. It is relevant to show the activity of LSD1 inhibition in additional human adoptive T cell therapy models. A relevant model could be human CD19 CAR-T cells, among others.

Response: Following the reviewer's suggestion, we have investigated the antitumor activity of LSD1 inhibition in animal models of human CAR T therapy. The results showed that LSD1 inhibition by GSK2879552 (abbreviated as GSK) strongly enhanced the antitumor effect of human CD19-CAR T cells against CD19-positive Nalm6 leukemia and A375 melanoma (transduced with CD19) (new **Fig. 7h-j, n**), in line with the findings from the OT-1 transfer model.

2. The field is discovering several similar approaches to increase the antitumor activity of adoptively transferred T cells based on epigenetic modifications. These approaches lead to similarly reported increases in T cell antitumor activity; however, it would be very relevant for the field to start benchmarking these approaches against each other. I would suggest comparing the epigenetic modifications and the antitumor activity obtained with LSD1 inhibitors and other epigenetic modulators such as DNMT3a inhibitors, decitabine, and EZH2 and SUV39H1 inhibitors.

Response: These are very constructive comments. To provide benchmarking for epigenetic approaches in adoptive T cell therapy, we have compared LSD1 inhibitors (GSK and ORY1001) with several reported epigenetic modulators, including decitabine (a DNMT inhibitor), GSK126 (an EZH2 inhibitor), and chaetocin (a SUV39H1 inhibitor), in terms of their effects on T cell activation and antitumor activity. While GSK and ORY1001 showed no dose-limiting toxicities for CD8⁺ T cells in the examined range of 50–1000 nM, decitabine and chaetocin caused severe cell

death at concentrations exceeding 50 nM or 20 nM, respectively. At lower doses, decitabine had a dose-dependent effect of upregulating PD-1 but downregulating SLAMF6 and CD62L, contrasting the observations with chaetocin and LSD1 inhibitors (new **Supplementary Fig. 2a-d**). GSK126 at 1 μ M concentration appeared to enhance the effector phenotype of T cells, marked by increased expression of PD-1, CD44, and GzmB (new **Supplementary Fig. 2a-f**). In contrast to other modulators, LSD1 inhibitors did not reduced Ki-67 expression in CD8⁺ T cells (new **Supplementary Fig. 2g**), and modestly lowered apoptosis levels of activated CD8⁺ T cells (new **Supplementary Fig. 2h, i**). These results suggest that LSD1 inhibitors outperform other epigenetic modulators by enhancing the memory/progenitor-associated phenotype and conferring a survival advantage to activated CD8⁺ T cells. Correspondingly, the potency of GSK priming in amplifying the antitumor efficacy of OT1 cells was on par with that of chaetocin priming but stronger than that of decitabine and GSK126 priming (**Supplementary Fig. 7a, b**). These findings highlight the effectiveness of LSD1 inhibition, representing a promising approach to improving adoptive T cell therapy.

3. A potential concern of this treatment is the antigen-independent release of cytokine that could lead to severe toxicities of this treatment. In Figure 1 (murine T cells) and Figure 7 (human T cells), T cells are activated for 48h and then expanded with and without drug and IL2. After expansion, these cells show higher levels of cytokines and granzyme without antigen stimulation that could lead to toxicities once administered systemically. Please expand on this issue and justify if this is not a concern. Toxicology data in vivo would be required to confirm the potential lack of toxicity.

Response: We thank the reviewer for raising the toxicity issue. To address this, we transferred 5 million Veh- or GSK-primed OT-1 cells to B16-OVA tumor-bearing mice for the assessment of toxicities. Over two weeks after OT-1 transfer, neither group of mice showed apparent weight loss (new **Supplementary Fig. 7c**). No increase in systemic levels of IL-6 or IL-1 β , two cytokines linked to cytokine release syndrome and neurotoxicity (PMID: 29808007, 29808005), was detected in any group of recipient mice (new **Supplementary Fig. 7d, e**). In addition, no evident lesions or immune cell infiltration were spotted in the histological sections of the brain, heart, or liver (new **Supplementary Fig. 7f**). Moreover, infusion of either control or GSK-primed CD19-CAR T cells to tumor-bearing NCG mice did not cause severe body weight loss, although Nalm6-bearing mice lost weight at a later phase due to disease progression (**Supplementary Fig. 7g, h**). Collectively, these data suggest that GSK2879552 priming enhances the antitumor efficacy of adoptively transferred T cells without leading to toxicities.

Minor comments:

Figure 2a. The lower part (Defined enhancers) is missing all the titles. It would be easier to read if added, like in the top part of the figure.

Response: Sorry for the omission. We have added all the titles in the lower part in the revised manuscript.

Line 334. I would suggest using “not statistically significant” instead of “statistically insignificant”.

Response: We took this suggestion and used “not statistically significant” in the revised manuscript.

Reviewer #2 (Remarks to the Author): with expertise in cancer, epigenetics, LSD1

This is a strong paper showing that transient LSD1 inhibition can augment adoptive T cell therapy. Most experiments are conducted in mouse T cells models and include melanoma tumor bearing models, and a final experiment uses human T cells in a Nalm6 model. Some data is overstated and requires additional validation as listed below:

1. Rationale for the dose of GSK2879552 and ORY1001 used in Figure 1 and supplementary Figure 1 should be provided. In particular, authors should show whether LSD1 is actually inhibited in T cells at these doses. This could be accomplished by CETSA.

Response: Thanks for the reviewer's critiques and suggestions. To clarify the rationale for the dose of GSK2879552 and ORY1001 used, we analyzed the phenotypes of CD8⁺ T cells in response to both inhibitors at a range of concentrations (50–1000 nM). The results showed that a concentration of 0.5 μM of either inhibitor was adequate to produce a maximum effect on the expression of cell surface markers, including PD-1, SLAMF6, and CD62L (new **Supplementary Fig. 2a-d**). Notably, this dose of inhibitors had no adverse effects on T cell survival or proliferation (new **Supplementary Fig. 2g, i**). As suggested, we performed the CETSA assay with serially diluted concentrations of inhibitors, which demonstrated the full target engagement and presumably complete inhibition of LSD1 in T cells by either inhibitor at 0.5 μM (new **Supplementary Fig. 1a-c**). Collectively, these results provide the rationale for the dose of GSK2879552 and ORY1001 used in this study.

2. Supplementary Figure 2a should be carried out with ORY1001 to confirm the survival advantage of activated CD8⁺ cells is seen with another inhibitor.

Response: As suggested, we have performed the CFSE cell proliferation assay with ORY1001, and observed that it modestly increased the proportion of T cells undergoing five rounds of cell division as did GSK2879552 (new **Supplementary Fig. 3a**).

3. Some explanation for the finding in Figure 2a should be offered regarding why H3K4me1 and H3K4me2 are unchanged. What are the putative genes that LSD1 is indirectly regulating?

Response: Thanks for the reviewer's comments. The observation that H3K4me1 and H3K4me2 demethylation by LSD1 only occurred at a small subset of LSD1-bound promoters and enhancers is consistent with previous reports (PMID: 29590629, 29590629), implicating a potential scaffold function of LSD1 in gene regulation. The interaction with transcription factors, for example Oct4 (PMID: 32023463), has been proposed to inhibit the histone demethylation activity of LSD1, which

might be an explanation for the unchanged H3K4me1 and H3K4me2 at LSD1 binding sites. Differentially expressed genes without LSD1 binding peaks at their promoters were considered to be indirectly regulated by LSD1 in this study. A list of those genes has been provided in a separate excel file (Please see Table 1 for Reviewers only). We have included this discussion in the revised manuscript.

4. Figure 3 and supplementary figure 4 should show EOMES depletion by western or flow.

Response: Flow cytometry data confirming EOMES depletion have been included in the revised manuscript (new **Supplementary Fig. 5c**).

5. A major finding of this study is in Figure 5 - showing that GSK priming duration is critical for efficacy. Can the authors correlate this with degree of kinetics and extent of LSD1 inhibition.

Response: We appreciate the reviewer's comment and suggestion. Our existing data underscore that the period of TCR stimulation is crucial for the efficacy of LSD1 inhibition. To evaluate the impact of GSK priming duration on the antitumor potency of adoptively transferred T cells, we administered GSK to *in vitro* activated OT1 cells for 2, 4, and 6 days, respectively, before transferring them to tumor-bearing mice after 6-days of total culture. Remarkably, 2-day GSK priming during TCR stimulation greatly improved the antitumor efficacy of OT1 cells compared with vehicle-treated counterparts (new **Fig. 5m**). Extending GSK priming to 6 days yielded incremental but less marked enhancements in efficacy (new **Fig. 5m**). Using the CETSA assay, we confirmed that the initial addition of 0.5 μ M GSK completely inhibited LSD1 during 2-day TCR stimulation period; removing GSK subsequently during the IL-2 expansion phase released the inhibition (new **Fig. 5o, p**). Together, these results suggest that 2-day GSK priming concurrent with TCR stimulation is the key parameter and, to a large extent, sufficient for LSD1 inhibition to potentiate the antitumor efficacy of OT1 cells.

6. Similar question for figure 6 - what is a molecular readout of adequate LSD1 priming?

Response: As discussed above, we found that 2-day GSK priming during TCR stimulation was largely adequate for the antitumor efficacy. 6-day GSK priming performed only modestly better than 2-day GSK priming. However, at the molecular level, the magnitude of change in gene expression (e.g., PD-1, SLAMF6, and CD62L) induced by 6-day GSK priming was much higher than that by 2-day priming (new **Fig. 5n**). This indicates that the antitumor efficacy of GSK priming does not directly correlate with the fold changes in gene expression. We agree that identifying a

molecular readout of adequate GSK priming will be informative, which is not trivial and warrants future investigation.

7. In Figure 7, why was the NALM6 model used for the human CAR T cells and not a melanoma model as done in the prior mouse expts?

Response: CAR T therapy has demonstrated clinical success in hematological malignancies but not in solid tumors yet, so we originally wanted to evaluate the translational significance of our findings by using a human leukemia model (Nalm6). Nevertheless, we have included the A375 melanoma model (transduced with CD19) in the revised manuscript, in which GSK-primed human CD19-CAR T cells consistently exhibited an enhanced antitumor efficacy (new **Fig. 7n**).

Reviewer #3 (Remarks to the Author): with expertise in cancer immunology

The manuscript by Qiu et al combines chemical inhibitor and genetic knockout data to reveal the potential role and relevant mechanism of LSD1 in suppressing adoptive CD8⁺T cell therapy. In particular, the authors report that the inhibition of LSD1 reshapes the epigenome and improves the memory phenotype of both murine and human CD8⁺ T cell, thus enhancing the antitumor effect in solid tumor models after adoptive transfer alone or in combination with PD1 blockade. Collectively, this is an interesting and comprehensive study that increases our understanding of T cell exhaustion regulation, especially PD1 regulation, and provide evidence for combined trial of LSD1 inhibition and PD1 blockade in the context of adoptive CD8⁺ T cell therapy. However, not all proposed mechanisms are sufficiently supported by the data.

Major comments:

1. For Fig. 1j, besides the cell death maker 7AAD, the authors should also check the expression of the proliferation maker Ki67.

Response: According to the reviewer's suggestion, we examined Ki-67 expression by flow cytometry, which showed that its expression in activated CD8⁺ T cells was not affected by 0.5 μ M GSK2879552 (abbreviated as GSK) but reduced by 50 nM decitabine (new **Supplementary Fig. 2g**).

2. The data in Fig. 3a-3b does not support the conclusion in line 183 that, GSK treatment inhibits TCR activation-induced PD1 expression. Please revise and do not over-interpret data.

Response: Thanks for pointing this out. We agree with the reviewer's comment, so we have removed that sentence in the revised manuscript.

3. For Fig. 5j-5k, the authors should also add a "GSK+IgG" group to demonstrate the tumor suppression effect of "GSK+anti-PD1" is superior to GSK treatment alone.

Response: We thank the reviewer for this critique. We have repeated this experiment with a full set of six groups (PBS + Isotype, OT1 Veh + Isotype, OT1 GSK + Isotype, PBS + α -PD-1, OT1 Veh + α -PD-1, and OT1 GSK + α -PD-1). The result showed that "OT1 GSK + α -PD-1" was superior to "OT1 GSK + Isotype" treatment at inhibiting B16-OVA tumor growth (new **Fig. 5j, k**).

4. Since there are a lot of grammar errors and misuse of tense in the manuscript, the authors had better ask for help from a professional or a native English speaker to correct the errors.

Response: We apologize for the grammar errors. Following the reviewer's advice, we have sought help from our colleague Dante Neculai for grammar correction. We hope the reviewer will find the revised manuscript satisfactory.

Minor comments:

1. For representative data in Fig. 1f, the authors should show the frequency and the way they set the gates for IL-2+, TNFa+ and IFNr+ cells.

Response: The gating and frequencies of cytokine-producing cells have been added in Fig. 1e (the previous Fig. 1f) in the revised manuscript.

2. For representative data in Fig. 4j-4l, the authors should mark the treatment type accordingly.

Response: Sorry for the omission. Treatment types have been marked in the representative flow plots in the revised manuscript.

REVIEWERS' COMMENTS

Reviewer #1 (Remarks to the Author):

Thank you for the revised version of the manuscript. I do not have further comments.

Reviewer #2 (Remarks to the Author):

The authors were responsive to critiques. Notably, they have carried out several new experiments to address LSD1 binding by drug in T cells using CETSA. Benchmarking to other epigenetically targeted agents is now addressed. Expansion to additional tumor models is also now included. I find this study to be suitable for publication in Nature Communications.

Reviewer #3 (Remarks to the Author):

The authors have addressed all my concerns. I have no more question.